# Learning Interestingness in Automated Mathematical Theory Formation

**George Tsoukalas**
UT Austin
george.tsoukalas@utexas.edu

**Rahul Saha**
UT Austin
rahul.saha@utexas.edu

**Amitayush Thakur**
UT Austin
amitayush@utexas.edu

**Sabrina Reguyal**
Princeton University, Stanford University
sreguyal@stanford.edu

**Swarat Chaudhuri**
UT Austin
swarat@cs.utexas.edu

## Abstract

We take two key steps in automating the open-ended discovery of new mathematical theories, a grand challenge in artificial intelligence. First, we introduce FERMAT, a reinforcement learning (RL) environment that models concept discovery and theorem-proving using a set of symbolic actions, opening up a range of RL problems relevant to theory discovery. Second, we explore a specific problem through FERMAT: automatically scoring the *interestingness* of mathematical objects. We investigate evolutionary algorithms for synthesizing nontrivial interestingness measures. In particular, we introduce an LLM-based evolutionary algorithm that features function abstraction, leading to notable improvements in discovering elementary number theory and finite fields over hard-coded baselines. We open-source the FERMAT environment at `github.com/trishullab/Fermat`.

## 1 Introduction

AI researchers have dreamed of building an "automated mathematician" since the 1950s [29]. Such a system would allow human mathematicians to harness the vast processing capacity of computers to discover entirely new areas of mathematics [42]. An emerging body of work seeks to realize this dream using the tools of modern machine learning. In particular, the AI community has developed a wide range of systems that can prove formal theorems [13, 47] and search for programs discovering mathematical constructions [31, 40].

However, a key limitation of much of this research is that it is focused on solving predefined problems. Mathematicians develop theories through an *open-ended* process of defining new concepts, studying their properties, making conjectures, and proving or finding counterexamples. While some work [35] has offered systems that construct new problems in addition to solving them, there is currently no framework that supports the full theory-formation process, including, for example, the synthesis of new definitions in addition to problems.

A central challenge in this open-ended process is guiding the search. The space of possible definitions and conjectures is combinatorially vast, and most paths lead to trivial or dull mathematics. Human mathematicians navigate this space using a nuanced, intuitive sense of "interestingness" — a judgment of scientific potential that directs their focus. An explicit formulation of this concept has long been debated, with different perspectives valuing properties such as the surprising connection between disparate fields [36], depth and generality [22], or its unexpected real-world applicability [49].

In this paper, we take two key steps towards addressing these challenges. First, we provide a reinforcement learning (RL) framework, called FERMAT (Figure 1), which can be used to design and

39th Conference on Neural Information Processing Systems (NeurIPS 2025).

evaluate new algorithms for automatic theory formation. The system generalizes the early symbolic computing-prover system HR [8], which used a system of *production rules* to generate new concepts and conjectures, either symbolically or from explicit examples, and *proof mechanisms* for resolving conjectures. We model these symbolic steps as the actions of a Markov Decision Process (MDP), and the mathematical knowledge available at a given point during exploration as an MDP state. This formulation opens up numerous RL problems relevant to theory formation.

Our second contribution is a solution to a particular algorithmic problem in FERMAT: learning an interestingness heuristic for selecting mathematical concepts to develop. To form a theory, one must navigate a combinatorial search space of mathematical objects, most objects in which are not meaningful or worthy of study. Prior works were attentive to this problem, but required hard-coded measures to formalize the concept of interestingness [8, 26]. In contrast, we frame the discovery of this heuristic as a learning problem. We specifically learn interestingness measures as *programmatic* representations, as this makes them interpretable and allows us to analyze what may contribute to fruitful discovery. To this end, we develop an LLM-driven method, called EvoAbstract, for learning the intrinsic value of mathematical objects in the context of the current theory. EvoAbstract is an evolutionary program synthesis algorithm that extends the FunSearch [40] approach with a form of abstraction learning, allowing for interpretable abstractions to be discovered during function search. We experimentally show that EvoAbstract can automatically synthesize interestingness measures that lead to significant improvements in discovering concepts in elementary number theory and finite fields over hard-coded baselines.

## 2 Problem Formulation and Motivation

### 2.1 Mathematical Theory Formation as a Markov Decision Process (MDP)

To rigorously study automated mathematical theory formation using reinforcement learning, we first formalize the process as an MDP $(\mathcal{S}, \mathcal{A}, \mathcal{T}, \mathcal{R})$. This framework allows us to model the sequential nature of mathematical discovery, where an agent iteratively expands a body of knowledge by making choices about definitions, conjectures, and proof attempts. Let $\mathcal{M}$ denote the universe of all well-formed mathematical entities. The components of this MDP are defined as follows:

Figure 1: A high-level description of FERMAT, our environment for mathematical theory formation. At any given time, the current theory (state) is represented as a knowledge graph consisting of the mathematical definitions, conjectures, and theorems discovered so far. At each step, the policy $\pi$ inputs the current state and selects an action to apply, updating the theory with additional information. The action space allows the production of new definitions, conjectures, and proofs of theorems.

• **Mathematical State Space** ($\mathcal{S}$): A state $S \in \mathcal{S}$ represents the current state of mathematical knowledge, represented as a *directed knowledge graph* $G = (V, E)$, where:

  - $V \subseteq \mathcal{M}$ is the set of **mathematical entities**, categorized into definitions $\mathcal{D}$, conjectures $\mathcal{C}$, and theorems $\mathcal{T}$.
  - $E$ is the set of dependency **edges**, where an edge $(u, v)$ exists if entity $u$ was used as direct input for the action that generated entity $v$, and is labeled with that action.

• **Action Space** ($\mathcal{A}$): An action $a \in \mathcal{A}$ represents an operation that modifies the knowledge graph by introducing a new entity or acting upon existing ones. Actions fall into the following categories:

  - **Definition Production Actions** ($\mathcal{A}_{def}$): Introduces a new definition, adding a node $d'$ to $G$ and connecting it to relevant entities via a function $\delta_{def} : \mathcal{S} \times \mathcal{A}_{def} \to \mathcal{S}$.
  - **Conjecture Production Actions** ($\mathcal{A}_{conj}$): Formulates a new conjecture $c'$ based on existing entities and relationships, governed by a function $\delta_{conj} : \mathcal{S} \times \mathcal{A}_{conj} \to \mathcal{S}$.
  - **Proof Actions** ($\mathcal{A}_{proof} = \{$***prove, disprove***$\}$): Verifies or refutes a conjecture $c \in \mathcal{C}$ by invoking a backend theorem prover, updating its status to theorem or disproven.

- **Transition Function ($\mathcal{T}$):** The transition function $\mathcal{T} : \mathcal{S} \times \mathcal{A} \times \mathcal{S} \to [0, 1]$ models how applying an action updates the knowledge graph. $\mathcal{T}(S, a, S')$ denotes the probability of transitioning from state $S$ to $S'$ after applying action $a$. In particular,

  - Adding a new definition or conjecture $c$ extends $V$ and introduces edges emanating from the entities to which the production rule was applied: $V' = V \cup \{c\}$, $E' = E \cup \{(v, c) \mid v \in V_{inputs}\}$.
  - A successful prove action converts a conjecture $c$ into a theorem and attaches a proof attribute: $\mathcal{C}' = \mathcal{C} \setminus \{c\}$, $\mathscr{T}' = \mathscr{T} \cup \{t\}$ with proof structure $\pi_t$. A successful disprove action refutes the conjecture $c$, marking it as false and attaching a counterexample as a witness, where possible.

- **Reward Function:** We design an extrinsic reward function $\mathcal{R}_{\mathcal{E}} : \mathcal{S} \times \mathcal{A} \times \mathcal{S} \to \mathbb{R}$ to incentivize the discovery of a pre-defined set $\mathcal{E}$ of well-known mathematical entities. Let the application of action $a$ to state $S$ produce a state $S'$ with a new entity $m_{new} \in \mathcal{M}$. The reward is defined as:

$$\mathcal{R}_{\mathcal{E}}(S, a, S') = \begin{cases} 1 & \text{if } m_{new} \in \mathcal{E} \\ 0 & \text{otherwise} \end{cases}$$

A reward is thus granted only when the agent's action results in the discovery of a specific ground-truth concept. A *policy*, denoted by $\pi(a|s)$, defines a strategy by specifying the probability of taking action $a$ in a given state $s$. A *rollout* refers to a single episode of interaction used to evaluate this policy by generating a *trajectory*, $\tau = (S_0, a_0, r_1, S_1, a_1, r_2, \ldots, a_{T-1}, r_T, S_T)$. This sequence is formed by starting in $S_0$ and repeatedly sampling an action $a_t \sim \pi(\cdot|S_t)$, after which the environment dictates the next state $S_{t+1} \sim \mathcal{T}(S_t, a_t, \cdot)$ and the corresponding reward $r_{t+1} = \mathcal{R}_{\mathcal{E}}(S_t, a_t, S_{t+1})$.

The *intrinsic reward* $\mathcal{R}_{\mathcal{I}}$ is a function $\mathcal{R}_{\mathcal{I}} : \mathcal{S} \times \mathcal{A} \times \mathcal{S} \to \mathbb{R}$ that serves as a mechanism for the agent/policy to learn effectively in a sparse extrinsic reward setting. Such internal rewards can be critical for driving exploration and acquisition of general knowledge about the environment and discovery of useful subgoals, especially when external feedback is infrequent or absent, by promoting behaviors like curiosity or novelty-seeking [1, 3, 32, 33, 41, 43].

## 2.2 Interestingness as Intrinsic Reward

Humans use intuition and are intrinsically motivated to define interesting mathematical goals. Capturing this notion for an autonomous agent is a key challenge. We approach this by modeling interestingness as a learnable intrinsic reward, guiding a policy to discover meaningful theory. In this work, we wish to discover such interestingness measures autonomously, and view the synthesis of an effective interestingness measure as a problem of *intrinsic reward optimization*.

Formally, we define the interestingness measure to be a function $\mathcal{I} : \mathcal{M} \times \mathcal{S} \to \mathbb{R}$, where $\mathcal{I}(m, S)$ provides an estimate of the value of a mathematical entity $m$ in the context of the current theory $S$. We connect this entity-scoring function to our RL framework by defining the intrinsic reward $\mathcal{R}_{\mathcal{I}}$ for a state transition as the interestingness score of the newly generated entity. If taking action $a$ in state $S$ produces a new entity $m_{\text{new}}$ in state $S'$, the intrinsic reward is $\mathcal{R}_{\mathcal{I}}(S, a, S') = \mathcal{I}(m_{\text{new}}, S')$.

In this work, we search over the class of interestingness measures that can be represented as programmatic functions. The policy, $\pi_{\mathcal{I}}$, is built using the measure $\mathcal{I}$ according to a fixed template (detailed in Section 5), and is designed to leverage the function's scores to guide its selection of actions. While the policy $\pi_{\mathcal{I}}$ acts based on this local, short-term measure, our learning problem is to discover an optimal function $\mathcal{I}^*$ that maximizes the cumulative long-term *extrinsic* reward:

$$\mathcal{I}^* = \arg\max_{\mathcal{I}} \mathbb{E}_{\tau \sim \pi_{\mathcal{I}}} \left[ \sum_t \gamma^t \mathcal{R}_{\mathcal{E}}(S_t, a_t) \right]$$

where $\gamma \in [0, 1]$ represents the discount factor. In Section 4, we detail our evolutionary algorithm designed towards discovering an optimal $\mathcal{I}$.

## 3 FERMAT: A Framework for Automated Theory Formation

In this section, we discuss FERMAT, our framework for automated mathematical theory formation built atop our MDP formulation of the theory discovery.

## 3.1 Mathematical Entities

The FERMAT framework, implemented in Python, provides the environment for automated theory formation. It is built upon a structured representation of mathematical entities within an evolving knowledge graph. At its core is a formal domain specific language (DSL) to define these entities.

Each mathematical entity $m$ within the knowledge graph $G = (V, E)$ (where $m \in V$) encapsulates its meaning through several key components:

(1) **Symbolic Definition** ($m_{sym}$). This holds the formal representation of the entity $m$ expressed in FERMAT's DSL. It precisely defines the entity's logical structure. For an entity $m = \texttt{is\_prime}$, the symbolic definition might be the following expressed programmatically,

$$m_{sym} = \ \lambda \texttt{n.} \ \big(\texttt{n} > \texttt{1}\big) \wedge \forall \texttt{k} \in \mathbb{N} \ . \ \Big( \underbrace{\exists \texttt{q} \in \mathbb{N} \ . \ \texttt{n} = \texttt{q} \times \texttt{k}}_{\texttt{divides(k,n)}} \ \Rightarrow \ (\texttt{k} = \texttt{1} \ \vee \ \texttt{k} = \texttt{n}) \Big)$$

(2) **Computational Interpretation** ($m_{comp}$). This is an executable Python function that provides an efficient, concrete evaluation of the entity's symbolic definition $m_{sym}$. Let $I$ be the space of potential input instances for $m$. The interpretation is a mapping $m_{comp} : I \to \{\text{True}, \text{False}, \text{Unknown}\}$ where, for an instance $i \in I$, the function returns:

- **True** (resp. **False**) if $i$ has been verified computationally (resp. not) to satisfy $m_{sym}$.
- **Unknown** when the evaluation of $i$ could not be determined computationally within resource limits (e.g. due to universal quantification over an infinite set).

As an example, for an entity $m = \text{square}$, its computational interpretation could be given by $m_{comp} = \texttt{lambda a, b: b == a*a}$.

(3) **Cached Instances** $\mathcal{X}(m) = (\mathcal{X}^+(m), \mathcal{X}^-(m))$. These components store explicit input instances for the entity $m$, where $\mathcal{X}^+(m) = \{i \in I \mid m(i) = \text{True}\}$ stores examples, and $\mathcal{X}^-(m) = \{i \in I \mid m(i) = \text{False}\}$ stores nonexamples. These instances ground the entity's semantics and can be used for various purposes. For $m = \text{divides}$:

$$\mathcal{X}^+(m) = \{(2,4), (1,3), (2,2), (3,6), \dots\}, \qquad \mathcal{X}^-(m) = \{(2,3), (3,5), (4,1), (5,4), \dots\}$$

We write $m_i, m_o$ for input and output arity of $m$, and $\text{size}(m) = m_i + m_o$ for size of the examples.

The construction history $\mathcal{C}(m) = \{a_1, \dots, a_n\}$ of an entity $m$ is the ordered list of actions applied to produce it. Definitions ($m \in \mathcal{D}$) are further classified as either **predicates** or **functions**. This informs which production rules in the action space $\mathcal{A}$ are applicable.

## 3.2 Production Rules

Following HR [8], FERMAT comes equipped with a set of *production rules*, consisting of composable actions for constructing new definitions and conjectures from prior ones. These rules define all construction actions in $\mathcal{A}_{def}, \mathcal{A}_{conj} \subseteq \mathcal{A}$ to produce new entities. We divide the production rules by whether they produce definitions or conjectures. We include a complete description of all the production rules present in FERMAT in Appendix A.1, and give two condensed examples:

**Definition Production Rules.**

(1) **Exists**: Let $\mathcal{P}(\mathbf{x}_1, \dots \mathbf{x}_n)$ be a predicate and $I := \{i_1, \dots, i_k\}$ be a list of indices to existentially quantify over, and let $J := \{j_1, \dots, j_{n-k}\}$ be the remaining indices in increasing order. Then the production rule outputs a new predicate $\mathcal{Q}$ as follows,

$$\texttt{exists} \ \mathcal{P} \ I \to \boxed{\mathcal{Q}(\mathbf{x}_{j_1}, \dots, \mathbf{x}_{j_{n-k}}) := \exists \mathbf{x}_{i_1}, \ \dots \ , \mathbf{x}_{i_k} \quad \text{s.t.} \quad \mathcal{P}(\mathbf{x}_1, \dots, \mathbf{x}_n)}$$

*Example.* Consider the predicate $P(x, y)$ defined by $P(x, y) :\iff \exists z., y = x \times z$, which expresses that $x$ divides $y$. This can be constructed by applying the *Exists* production rule to the predicate $\text{product}(x, z, y)$, which holds when $y = x \times z$, with the index list $I = \{1\}$ corresponding to the variable $z$ to be existentially quantified. Formally,

$$\texttt{exists} \ \text{product} \ I \longrightarrow \boxed{Q(x, y) := \exists z \ \text{s.t.} \ \text{product}(x, z, y)}$$

(2) **Specialize**: Given an entity, this rule outputs a new definition by specializing a variable to a fixed value. Let $\mathcal{A}(\mathbf{x}_1, \ldots, \mathbf{x}_n)$ be a function (resp. predicate), and let $i$ be the index to specialize, and $v$ be the value to substitute. Then the rule outputs a function (resp. predicate) $\mathcal{B}$ as follows,

$$\texttt{specialize } \mathcal{A}\ i\ v \rightarrow \boxed{\mathcal{B}(\mathbf{x}_1, \ldots, \mathbf{x}_{i-1}, \mathbf{x}_{i+1}, \ldots, \mathbf{x}_n) := \mathcal{A}(\mathbf{x}_1, \ldots, \mathbf{x}_{i-1}, v, \mathbf{x}_{i+1}, \ldots, \mathbf{x}_n)}$$

FERMAT contains 7 other definition production rules: (i) *Compose*, which composes definitions, (ii) *MapIterate*, which successively applies an iterator function, (iii) *ForAll*, which universally quantifies over variables in definitions, (iv) *Match*, which asserts equality of chosen variables in definitions, (v) *Negate*, which outputs the negation of a concept, (vi) *Size*, which outputs a definition of the cardinality of the set of inputs satisfying a condition, and (vii) *Constant*, which creates constants from examples.

**Conjecture Production Rules.** FERMAT contains 4 production rules designed to construct conjectures. These are: (i) *Implication*, which asserts that one definition implies another over all inputs, (ii) *Equivalence*, which asserts that two definitions are equivalent, (iii) *Nonexistence*, which asserts that no examples of a definition exist, (iv) *Exclusivity*, which asserts that the only examples of a given definition belong to a given finite set.

## 3.3  Prover

To complete the action space, we require the ability to validate conjectures generated using FERMAT's DSL. Critically, the generic DSL supports *nested definitions*, facilitating modular construction of definitions and conjectures. These conjectures, which may involve previously defined concepts, are automatically constructed by our framework and passed to a backend theorem prover for verification. We instantiate this backend using the Z3 Theorem Prover [11], and provide it with SMT-LIB input generated from our DSL via a custom-designed compiler. We choose Z3 as it is a powerful off-the-shelf black-box prover, the use of which enables us isolate the problem of synthesizing definitions and conjectures. We include examples of the Z3 support available through our DSL, and compilation down to SMT-LIB format in Appendix A.3.

## 4  Learning Interestingness

In this section, we discuss our approach for automatically *learning* an interestingness measure $\mathcal{I}(m, S)$ that guides the agent in discovering human mathematical knowledge. Following HR [8], which developed simple programmatic representations of measures over features of the state, we *search* in the space of Python programs that implement interestingness measures. To this end, we introduce EvoAbstract (Figure 2), an evolutionary search algorithm designed to optimize an objective function given a simple numerical evaluator function.

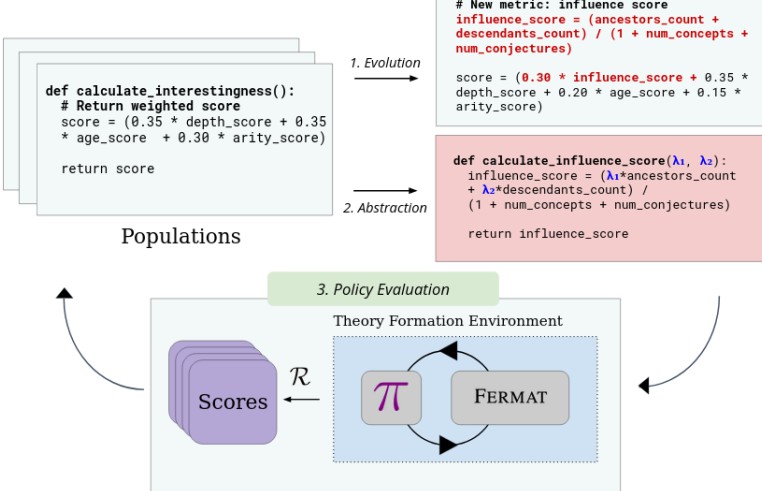

**LLM-Driven Evolutionary Search.** (EVOLUTIONSTEP). At its core, EvoAbstract is an evolutionary algorithm, aimed at synthesizing programs iteratively. Each population consists of candidate interestingness programs. The generation of

Figure 2: Overview of EvoAbstract, which aims to discover an optimal interestingness measure for mathematical theory formation. It operates through three phases: (1) **Evolution**, where populations of candidate programs are generated and refined through LLM-driven mutations; (2) **Abstraction**, where high-performing programs are analyzed and reusable subroutines are extracted; and (3) **Policy Evaluation**, where the resulting programs are evaluated within the theory formation environment using FERMAT, producing feedback that guides subsequent evolutionary steps.

new programs is primarily driven by an LLM, $\mathcal{L}_{var}$, conditioned on a prompt instructing it to perform evolution. In each evolutionary step, $\mathcal{L}_{var}$ takes the program template $T$ and a selection of high-performing parent programs from a population as input, and synthesizes new candidate solutions ($f_{new}$). $\mathcal{L}_{var}$ thus acts as an operator for exploration and exploitation, intended to perform complex mutations informed by successful prior programs. We employ an island model with $k$ parallel populations ($\mathcal{P}_i$) to maintain diversity.

**Abstraction Learning.** (ABSTRACTIONSTEP). Our central innovation in EvoAbstract is its abstraction learning mechanism. This component is designed to identify and reuse valuable subroutines from evolved programs. This system comprises two main parts:

- **Discovering and Utilizing Abstractions:** Periodically EvoAbstract enters an abstraction phase, where an LLM $\mathcal{L}_{abs}$, analyzes a set of high-scoring programs ($S'_i$) sampled from each population. $\mathcal{L}_{abs}$ is tasked with identifying *abstractions* — valuable, reusable subroutines with defined signatures and implementations —— within these successful programs and proposing them as new, generalized functions ($A_{new}$). These proposed abstractions are then filtered for criteria such as syntactic validity and uniqueness before being added to the island's Lib$_i$.
- **The Abstraction Library (Lib$_i$):** Each island $i$ maintains a dynamic *Abstraction Library*, Lib$_i$. This library serves as a repository for functional abstractions that are identified as potentially useful during the search. Initially, these libraries are empty. After each abstraction phase, the generated abstractions are added to their respective libraries Lib$_i$. The evolutionary LLM, $\mathcal{L}_{var}$, is conditioned not only on the template $T$ and sampled programs but, crucially, also given access to the current set of abstractions in Lib$_i$ when generating new candidate programs. This encourages $\mathcal{L}_{var}$ to compose solutions by utilizing these validated sub-components, thereby promoting modularity, facilitating the construction of more complex solutions, and guiding the search towards more promising regions of the program space.

**Policy Evaluation.** (POLICYEVALUATIONSTEP). In each iteration, candidate programs produced through evolution are assessed via episodic rollouts within the theory-formation environment. During a rollout, a policy instantiated by an interestingness program interacts with FERMAT to guide the discovery process over multiple steps. The cumulative reward obtained across these rollouts determines each program's fitness, providing the signal that drives subsequent evolutionary and abstraction phases.

The overall EvoAbstract algorithm, detailed in Algorithm 1 (Appendix), thus proceeds in generations. Within each generation, the evolutionary search driven by $\mathcal{L}_{var}$ refines the populations on each island. Periodically, the abstraction phase mediated by $\mathcal{L}_{abs}$ enriches the abstraction libraries, which in turn provide more powerful building blocks for subsequent evolutionary steps. This interplay between LLM-driven evolution and LLM-driven abstraction learning allows EvoAbstract to progressively discover and refine programmatic subroutines.

## 5 Experiments

In this section, we present empirical results evaluating the effectiveness of our approach. We aim to answer key questions about the ability of EvoAbstract to learn effective interestingness measures and the capability of FERMAT, guided by these measures, to generate meaningful mathematical theories.

**Environment Configuration.** FERMAT centrally supports exploration in elementary number theory and finite fields, as these areas are extremely rich while easily represented. The number theory environment is supported by the Z3 Theorem Prover, while finite field reasoning is handled by a custom prover implemented in Python. For number theory, the ground truth benchmark $\mathcal{E}$ used for the extrinsic reward function $\mathcal{R}$ comprises 180 concepts, conjectures, and theorems sourced from an introductory number theory textbook [2], constituting a set of interesting entities. We similarly curated 67 such ground truth entities over $\mathbb{F}_{27}$, the primary finite-field setting in our experiments. These concepts span a range of theoretical sophistication, from the reflexive properties to the Goldbach conjecture. A detailed description of $\mathcal{E}$ along with subsets of the ground truth knowledge graph is detailed in Appendix A.4. For our experiments, we use three different starting configurations: (i) `succ_zero_eq` — The definitions of zero, successor function, and the equality predicate with arity 2; (ii) `arithmetic_base` — Containing zero, one, two, addition, multiplication, divides, $\leq$, and the

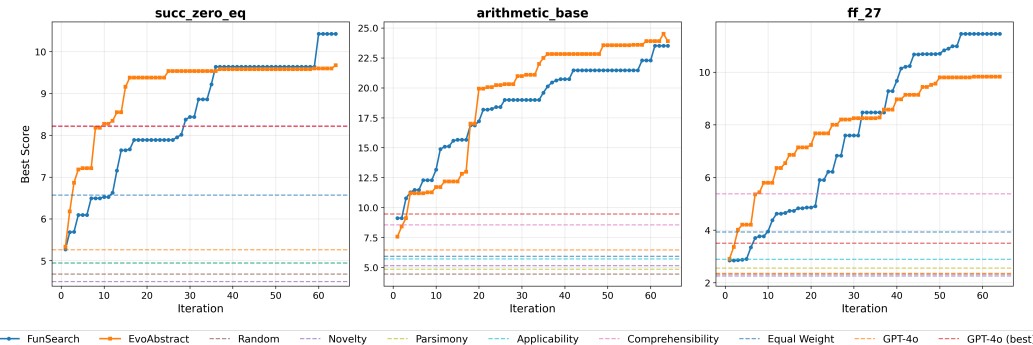

Figure 3: A plot of the best program found per iteration for FunSearch and EvoAbstract, shown for the three different starting knowledge graphs, averaged over four runs. As can be seen, the EvoAbstract and FunSearch methods dominate performance universally across all baselines. On `arithmetic_base`, EvoAbstract slightly outperforms FunSearch, while on `succ_zero_eq` and `ff_27` EvoAbstract optimizes the interestingness measures early on, but its performance plateaus sooner than FunSearch, which continues to improve.

equality predicate; (iii) `ff_27` — Defining zero, one, and generators of $\mathbb{F}_{27}$. We include the policy template in Algorithm 2 (Appendix).

**Evaluation Metrics.** To evaluate an interestingness measure, we instantiate the scoring function as extrinsic reward obtained through episodic rollouts of a policy depending on the measure through FERMAT. We run 64 episodes with a timeout of 60 seconds*, and average the reward. This evaluation metric measures the ability of the given policy to reconstruct the curriculum of human-made ground truth mathematical entities $\mathcal{E}$. We also provide qualitative analysis of the learned interestingness measures and the content of the generated theories.

**Baselines.** We compare EvoAbstract against the following baseline methods for generating or selecting interestingness measures:

- **Random Policy:** Selects applicable actions uniformly at random.
- **HR Measures:** We re-implement a number of interestingness measures manually defined in HR [8], which operate on the state $S$ and the newly generated entity $m$ (which can be extracted from the action $a$ applied and the new state $S'$). In particular, we include the following measures:
  1. *Novelty*. Computes the fraction of entities with the same example classification: $M_{\text{novelty}}(m) = \#\{m' \in S | \mathcal{X}(m) = \mathcal{X}(m')\}/\#S$.
  2. *Parsimony*. Rewards a concept with fewer inputs: $M_{\text{parsimony}}(m) = \text{size}(m)^{-1}$.
  3. *Productivity*. Measures how many subsequent environment steps use that entity in a production rule: $M_{\text{productivity}}(m) = \#\{m' \in S | m \text{ is in an action} \in \mathcal{C}(m')\}/\#S$.
  4. *Applicability*. Computes the fraction of all known instances that are examples: $M_{\text{applicability}}(m) = \mathcal{X}^+(m)/(\mathcal{X}^+(m) + \mathcal{X}^-(m))$.
  5. *Comprehensibility*. Rewards a concept which is more comprehensible, measured by the inverse of the number of construction steps: $M_{\text{comprehensibility}}(m) = \#\mathcal{C}(m)^{-1}$.

  We evaluate these measures individually and when combined in an equally weighted sum. Note that all of these measures are easily representable as Python programs.
- **One-shot LLM.** Instead of evolving a program, we sample 64 programs from GPT-4o and evaluate their performance through episodic roll-outs, averaging the result.
- **FunSearch:** An ablation study where EvoAbstract is missing the abstraction component, which is equivalent to the FunSearch [40] algorithm without island crossover, ran at a scale afforded by our budget. We use the same hyperparameters as in our EvoAbstract evaluation.

**EvoAbstract configuration.** We configure EvoAbstract to employ $k = 4$ islands and runs over $N_{gen} = 64$ iterations, with each interestingness function being evaluated in 16 i.i.d. rollouts. We

---

*We note that we run episodes for a duration rather than a step count due to high variance in the time taken for Z3 to resolve conjectures.

run every configuration of EvoAbstract & FunSearch 4 times and average the results. We instantiate both the evolution and abstraction samplers $\mathcal{L}_{var}, \mathcal{L}_{abs}$ to use GPT-4o-mini, and sample 2 programs per iteration. We perform the abstraction phase every 8 iterations, sampling at most two abstractions per island. $\mathcal{L}_{var}, \mathcal{L}_{abs}$ are conditioned through prompting: a system-level instruction on generating interestingness measures is attached, as well as a description of a set of Python functions which return features of the state's knowledge graph representation as well as individual entities. These functions represent the base features for the interestingness measures to manipulate. We provide a more detailed list of hyperparameters, the full prompts, and our computational resources for experiments in Appendix A.5.

## 5.1 Experimental Results

We address the following research questions:

**RQ1:** Can EvoAbstract effectively learn interestingness measures that outperform baseline strategies in discovering ground-truth mathematical entities?

**RQ2:** What do the learned interestingness measures look like? Do they capture non-trivial patterns?

**RQ3:** Can we rediscover well-known concepts in elementary number theory and finite fields?

**Performance Comparison (RQ1).** We compare the cumulative extrinsic reward of policies guided by measures from EvoAbstract against baselines, with results summarized in Table 4. As expected, starting with a larger initial theory (`arithmetic_base`) generally leads to greater rewards.

Among static HR measures, the random and novelty measures perform worst, exhibiting roughly equivalent scores. Parsimony's inefficacy likely stems from its limited discriminative power, as most generated definitions involve few inputs, offering insufficient signal. Comprehensibility is the strongest HR measure as it rewards simplicity of entities. However, it alone cannot scale to more complex entities due to the combinatorial expansion of the action space.

| Measure | succ_zero_eq | arithmetic _base | ff_27 |
|---|---|---|---|
| Random | 4.68 (2.25) | 4.44 (2.23) | 2.33 (1.20) |
| Novelty | 4.50 (2.39) | 5.14 (2.83) | 2.26 (1.47) |
| Parsimony | 4.94 (2.90) | 4.85 (3.09) | 2.56 (1.32) |
| Applicability | 4.95 (2.25) | 5.71 (3.05) | 2.89 (1.69) |
| Comprehensibility | 8.23 (2.84) | 8.55 (3.22) | 5.38 (1.89) |
| Equal Weight | 6.57 (2.45) | 5.93 (2.82) | 3.93 (2.82) |
| GPT-4o | 5.26 (1.11) | 6.46 (1.98) | 2.36 (0.40) |
| GPT-4o (best) | 8.21 (4.09) | 9.45 (3.44) | 3.50 (1.87) |
| FunSearch | 10.23 (1.70) | 22.41 (2.68) | 11.34 (4.09) |
| EvoAbstract | 9.62 (2.97) | 23.98 (10.50) | 9.82 (4.83) |

Figure 4: Performance comparison of EvoAbstract against various baseline measures on three starting theories: `succ_zero_eq`, `arithmetic_base`, and `ff_27`. Each baseline receives 64 theory-formation evaluations. For FunSearch and EvoAbstract, we include the average score (standard deviation) of the best found program over four independent runs.

Interestingly, the GPT-4o baseline performs only slighter better than even the random baseline (see Figure 14), and is outperformed by just the comprehensibility measure. Despite generating more complex measures, its emphasis on rewarding construction depth and connectivity often assigns disproportionately high interestingness to initial, but irrelevant, entities. This leads to a cascading effect away from the ground truth set $\mathcal{E}$, explaining its suboptimal performance.

FunSearch [40] & EvoAbstract demonstrate the value of evolutionary search. In contrast to GPT-4o, where few generated measures surpassed the random baseline, evolutionary program synthesis yields significantly more performant measures, with the best measure discovered averaging (10.23, 22.41, 11.34) ground-truth entities per episodic roll-out on (`succ_zero_eq`, `arithmetic_base`, `ff_27`). Incorporating the abstraction phase in EvoAbstract introduces slight gains on `arithmetic_base`, yielding measures that discover an average of 23.98 ground-truth entities, but with higher variance. Notably, on `ff_27` and `succ_zero_eq`, EvoAbstract finds better solutions quicker, but the progress slows down and yields suboptimal performance at the end of the runs, on average. The abstractions are helpful in optimizing on known patterns, but produces an abstraction "lock-in" later on where it is difficult for the LLM to produce diverse samples that continue to increase the reward. Beyond improved discovery, this phase also develops interpretable modular components. Figure 3 illustrates the performance trajectory of EvoAbstract & FunSearch compared to the baselines.

**Analysis of Learned Interestingness Measures (RQ2).** We also conduct a qualitative analysis of the measures that EvoAbstract synthesizes. Figures 15, 17 presents an example of the best-performing program discovered by EvoAbstract on the `succ_zero_eq` task. A key characteristic of this program is its utilization of numerous abstractions and subroutines that were identified and refined during earlier abstraction phases. These abstractions are detailed in Figures 16, 18, which we now analyze.

First, EvoAbstract rediscovers and often refines variants of the baseline HR measures. For instance, it generates applicability-like measures, such as `compute_example_balance`, which calculates the ratio of examples to nonexamples. Notably, EvoAbstract can refine previous abstractions, exemplified by `calculate_uniqueness_score_v2`, which generalizes prior uniqueness abstractions. Furthermore, measures distinct from the HR baselines are found, such as `calculate_rule_diversity_score`, which weighs the diversity of rules in the construction history. Additionally, it produces abstractions which generalize known construction patterns, as seen from `adjust_score_by_node_type`.

A comparison with the best program generated by FunSearch (detailed in Figure 19) is instructive. While the FunSearch program utilizes similar components to those found by EvoAbstract, they are fewer in number. FunSearch tends to integrate these functionalities more directly, resulting in a less modular structure. The distinct modularity evident in Figure 15 lends itself to quicker readability of the high-level operation of the interestingness function.

**Analysis of Generated Theories (RQ3).** EvoAbstract & FunSearch discover a notable portion of fundamental math concepts in our ground truth benchmark. When starting from the `succ_zero_eq` base, the agent successfully develops the notion of addition, multiplication, divisibility, and the tau function. Furthermore, it makes progress towards conjecturing fundamental properties of divisibility, such as the reflexivity of divisibility. When starting from `arithmetic_base`, the agent goes further — discovering the concepts of powers and primality along with more complex compositions of functions. In `ff_27`, the evolutionary methods are capable of discovering concepts such as `ff_sum_three_times`, but cannot find the conjecture stating the characteristic of $\text{char}(\mathbb{F}_{27}) = 3$, which requires further rule applications to discover. Relevant samples of the evolved knowledge graphs are shown in Figure 20.

We note that the best-performing interestingness measures we find can still be suboptimal, upweighting entities are not particularly interesting to humans. For instance, we find that `equals`, which important for initial exploration, is assigned overly high interestingness, leading to an excess of redundant or vacuous statements during theory formation. While the agent generated conjectures, it had difficulty discovering many *ground truth* conjectures, which is likely due to the limited correct ways to correctly specify a conjecture compared to a definition.

## 5.2 Discussion

We find that there are several avenues for further exploration towards discovering richer theories. Firstly, the policy template we employ, designed to manage combinatorial growth, exposes only a subset of complete action space at any step. This choice, while pragmatic, limits scalability to more complex mathematical objects where a lengthy list of actions must be applied in a specific order. Secondly, we observe that there are "bottleneck" entities, such as primality, which must be discovered in order to continue the development of an interesting theory (see Figure 8). In our experiments, when primality is discovered, the resultant knowledge graph is prohibitively large so as to obstruct valuable actions involving it. Finally, FERMAT does not yet exploit symmetries in entities leading to representational redundancy. For instance, while exhaustively checking for equivalences between definitions would reduce this redundancy, we found the approach to be computationally intractable with Z3 as theories grow. Further experimentation with FunSearch and EvoAbstract with heavy compute budgets will help to investigate the potential for significant discovery with evolutionary methods in these domains. Addressing these points will be crucial for advancing FERMAT's ability to construct deeper and more sophisticated mathematical theories.

## 6 Related Works

**Automated Theory Formation.** AM (1977) [26, 27] was a theory formation program which relied on a curated set of 243 heuristics to discover concepts and prove conjectures in elementary set theory and number theory. Similarly, the Graph Theorist (1987) [17] performed conjecturing &

proving using an input set of definitions. HR (2000) [8] introduces a small set of production rules and manually curated heuristics to perform mathematical theory formation. Theorema (2006) [6] performed human-in-the-loop theory exploration, leveraging computational tools in Mathematica and with an emphasis on producing human-readable proofs. Theory formation is less explored in the modern era. Notably, Minimo [35] trains a neural model to play a game of conjecture and proof, but remains restricted to the initial axiomatic definitions. QuickSpec [44] is a symbolic theory exploration tool that interleaves term generation and random testing for conjecturing. As a final note, automated theory formation can be studied for domains other than mathematics — BACON (1983) [25] represents a program which aimed to rediscover empirical laws in chemistry.

**Conjecturing.** Many works have focused on the particular problem of synthesizing plausible conjectures. The PSLQ algorithm [19] was developed for identifying integer relations between mathematical constants. Graffiti & TxGraffiti [18, 10] produced conjectures in graph theory using several heuristics, given a large set of graphs and graph invariants. The Ramanujan Machine [4] utilized several algorithms to conjecture relations between fundamental constants such as $\pi$ and $\zeta(3)$. Davies et al. [9] uses machine learning techniques to identify patterns that lead to conjectures.

**Theorem-Proving.** The most significant attention in modern research has been applied towards the problem of theorem-proving. Simon & Newell's Logic Theorist and Hao Wang's Program II [29, 48], were early explorations into a theorem-proving system. Recently, neural systems [15, 24, 38, 45, 37] invoking interactive theorem provers like Lean [12], Isabelle [34], and Coq [46] have seen great interest. AlphaProof [13] and AlphaGeometry [47] together attained a silver medal at the International Mathematical Olympiad. Another interesting angle for resolving conjectures comes from use of SAT & SMT solvers — notably, yielding a resolution to the Boolean Pythagorean Triples problem [23]. Notably, the Four Colour Theorem was proved through computer-assisted case-checking [39]. Similarly, [7] used a combination of neural and symbolic techniques to disprove conjectures.

**Program Synthesis & RL.** FunSearch [40] and AlphaEvolve[31] are LLM-guided evolutionary algorithms used to discover programs producing mathematical constructions. The closely related LaSR [21] algorithm uses LLM-guided evolution and a learned *textual* abstraction library for symbolic regression. Eureka [28] uses iterative LLM refinement to produce extrinsic reward functions that outperform human-engineered rewards on a suite of RL environments. Several efforts [5, 16, 20] perform program synthesis in functional languages and develop abstraction libraries using symbolic abstraction algorithms. An interesting direction would be to develop separate explorative and exploitative policies, as in [30], for mathematical theory formation.

# 7 Conclusion

In this work, we targeted the problem of capturing the interestingness of concepts in the context of open-ended mathematical discovery. To support our research, we introduced FERMAT, a novel RL environment for theory formation. We introduced EvoAbstract, an LLM-based evolutionary procedure which abstracts and stores useful subroutines identified during search. We show that our learned interestingness measures outperforms several baselines when conducting theory formation using FERMAT, starting from basic definitions, in elementary number theory and finite fields.

Our investigation is a starting point for much broader research in automated theory formation. Integrating interactive theorem provers, like Lean [12], into FERMAT will allow exploration in more complex domains and enable studying the problem of *learning to prove tabula rasa*. It is also an open problem how to autonomously synthesize production rules. In future work, we see that extensions of FERMAT could lead to the development of new mathematics as envisioned in the early days of Artificial Intelligence.

# 8 Acknowledgement

This research was supported in part by NSF awards CCF-2212559 and CCF-2403211, a grant from Renaissance Philanthropy's AI for Math Fund, and the NSF AI Institute for Foundations of Machine Learning. We would also like to thank the anonymous reviewers for their insightful feedback, which helped improve the quality of this manuscript.

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

# A Appendix / supplemental material

## A.1 Production Rules.

Here we include a full description of each production rule available in FERMAT, including those mentioned in the main content for completeness.

**Definition Production Rules.**

1. **Compose**: This rule allows for composition of functions, predicates, and function-to-predicates.

    **Function Composition**. Let

    $$\mathcal{F} : X_1 \times \ldots \times X_n \to Y_1 \times \ldots \times Y_m, \qquad \mathcal{G} : Z_1 \times \ldots \times Z_k \to W$$

    be functions, and $I : \{1, \ldots, m\} \to \{1, \ldots, k\}$ be a map from $\mathcal{F}$'s output indices to $\mathcal{G}$'s input indices. Let $\mathbf{x}_1, \ldots, \mathbf{x}_n$ denote the parameters to be passed into $\mathcal{F}$, and $\mathbf{p}_1, \ldots, \mathbf{p}_i$ denote any additional parameters to be passed to $\mathcal{G}$. Then the production rule outputs a function as follows,

    $$\texttt{compose } \mathcal{F} \, \mathcal{G} \, I \; \to \; \boxed{\mathcal{H}(\mathbf{x}_1, \ldots, \mathbf{x}_n, \mathbf{p}_1, \ldots, \mathbf{p}_i) := \mathcal{G}(\mathbf{v}_1, \ldots, \mathbf{v}_k)}$$

    where

    $$\mathbf{v}_j \begin{cases} = \mathcal{F}(\mathbf{x}_1, \ldots, \mathbf{x}_n)_i, & \text{if } j = I(i), \\ \in \{\mathbf{p}_1, \ldots, \mathbf{p}_i\}, & \text{if } j \notin \text{Image}(I). \end{cases}$$

    **Predicate Composition**. Let

    $$\mathcal{P} : X_1 \times \cdots \times X_n \longrightarrow \text{Bool}, \qquad \mathcal{Q} : Z_1 \times \cdots \times Z_k \longrightarrow \text{Bool}.$$

    be predicates, and $S : \{1, \ldots, n\} \to \{1, \ldots, k\}$ be a sharing map from $\mathcal{P}$'s input variables to $\mathcal{Q}$'s input variables, that is, $S(i) = j$ if the $i$-th input variable of $\mathcal{P}$ and $j$-th input variable of $\mathcal{Q}$ will be shared when constructing the output predicate $\mathcal{R}$ defined below. Let $\text{Image}(S) = \{i_1, \ldots, i_s\}$ with $i_1 < \ldots < i_s$. Then the production rule outputs a predicate as follows,

    $$\texttt{compose } \mathcal{P} \, \mathcal{Q} \, S \; \to \; \boxed{\mathcal{R}(\mathbf{x}_1, \ldots, \mathbf{x}_n, \, \mathbf{p}_1, \ldots, \mathbf{p}_i) := \mathcal{P}(\mathbf{x}_1, \ldots, \mathbf{x}_n) \, \wedge \, \mathcal{Q}(\mathbf{v}_1, \ldots, \mathbf{v}_k)}$$

    where

    $$\mathbf{v}_j \begin{cases} = \mathbf{x}_i, & \text{if } j = I(i), \\ \in \{\mathbf{p}_1, \ldots, \mathbf{p}_i\}, & \text{if } j \notin \text{Image}(I). \end{cases}$$

    **Function to Predicate Composition**. This case works identically to function composition. Let

    $$\mathcal{F} : X_1 \times \ldots \times X_n \to Y_1 \times \ldots \times Y_m, \qquad \mathcal{P} : Z_1 \times \ldots \times Z_k \to \text{Bool}$$

    be a function and a predicate, and $I : \{1, \ldots, m\} \to \{1, \ldots, k\}$ be a map from $\mathcal{F}$'s output indices to $\mathcal{P}$'s input indices. Let $\mathbf{x}_1, \ldots, \mathbf{x}_n$ denote the parameters to be passed into $\mathcal{F}$, and $\mathbf{p}_1, \ldots, \mathbf{p}_i$ denote any additional parameters to be passed to $\mathcal{P}$. Then the production rule outputs a predicate as follows,

    $$\texttt{compose } \mathcal{F} \, \mathcal{P} \, I \; \to \; \boxed{\mathcal{H}(\mathbf{x}_1, \ldots, \mathbf{x}_n, \mathbf{p}_1, \ldots, \mathbf{p}_i) := \mathcal{P}(\mathbf{v}_1, \ldots, \mathbf{v}_k)}$$

    where

    $$\mathbf{v}_j \begin{cases} = \mathcal{F}(\mathbf{x}_1, \ldots, \mathbf{x}_n)_i, & \text{if } j = I(i), \\ \in \{\mathbf{p}_1, \ldots, \mathbf{p}_i\}, & \text{if } j \notin \text{Image}(I). \end{cases}$$

2. **Exists**: This rule allows for existentially quantifying out variables in a predicate or function.

    **Predicate**. Let $\mathcal{P}(\mathbf{x}_1, \ldots \mathbf{x}_n)$ be a predicate, and let $I := \{i_1, \ldots, i_k\}$, where $k < n$, be a list of input indices to existentially quantify over. Let $J := \{1, \ldots, n\} \setminus I = \{j_1, \ldots, j_{n-k}\}$

with $j_1 < \ldots j_{n-k}$ be the remaining indices. Then the production rule outputs a new predicate $\mathcal{Q}$ as follows,

$$\texttt{exists } \mathcal{P} \, I \rightarrow \boxed{\mathcal{Q}(\mathbf{x}_{j_1}, \ldots, \mathbf{x}_{j_{n-k}}) := \exists \mathbf{x}_{i_1}, \, \ldots, \mathbf{x}_{i_k} \quad \text{s.t.} \quad \mathcal{P}(\mathbf{x}_1, \ldots, \mathbf{x}_n)}$$

**Function**. Let $\mathcal{F}(\mathbf{x}_1, \ldots \mathbf{x}_n)$ be a function, and let $I$ and $J$ be defined similarly as before. Then the production rules outputs a new predicate $\mathcal{Q}$ as follows,

$$\texttt{exists } \mathcal{F} \, I \rightarrow \boxed{\mathcal{Q}(\mathbf{x}_{j_1}, \ldots, \mathbf{x}_{j_{n-k}}, \mathbf{y}) := \exists \mathbf{x}_{i_1}, \, \ldots, \mathbf{x}_{i_k} \quad \text{s.t.} \quad \mathcal{F}(\mathbf{x}_1, \ldots, \mathbf{x}_n) = \mathbf{y}.}$$

3. **Map Iterate**: This rule turns an iterator function (unary or binary) into a new function by applying the iterator function $n$ times.

   **Unary Function**. Let $\mathcal{F}$ be a unary iterator function. Let $\mathcal{F}^n$ be the $n$-fold application of $\mathcal{F}$, that is, $\mathcal{F}^n(x) = \underbrace{\mathcal{F}(\mathcal{F}(\ldots \mathcal{F}(x))\ldots)}_{n \text{ times}}$. Then this rule outputs a new function $\mathcal{G}$ as follows,

$$\texttt{map\_iterate } \mathcal{F} \rightarrow \boxed{\mathcal{G}(\mathbf{x}, n) := \mathcal{F}^n(\mathbf{x})}.$$

   **Binary Function**. Let $\mathcal{F}$ be a binary iterator function, and $v$ be an initial value concept to be passed into the iterator. Then this production rule outputs a new function $\mathcal{G}$ as follows,

$$\texttt{map\_iterate } \mathcal{F} \, v \rightarrow \boxed{\begin{array}{l} \mathcal{G}(\mathbf{x}, 0) := v, \\ \mathcal{G}(\mathbf{x}, n+1) := \mathcal{F}(\mathcal{G}(\mathbf{x}, n), \mathbf{x}). \end{array}}$$

4. **Forall**: This rule allows for universal quantification of variables over either one or two predicates.

   **Single Predicate**. Let $\mathcal{P}(\mathbf{x}_1, \ldots, \mathbf{x}_n)$ be a predicate, and let $U = \{u_1, \ldots, u_j\}$ be a list of indices of the variables to universally quantify over. Let $\bar{U} = \{\bar{u}_1, \ldots, \bar{u}_{n-j}\}$ be the remaining indices of the free variables. The production rule outputs a new predicate $\mathcal{R}$ such that

$$\texttt{forall } \mathcal{P} \, U \rightarrow \boxed{\mathcal{R}(\mathbf{x}_{\bar{u}_1}, \ldots, \mathbf{x}_{\bar{u}_{n-j}}) := \quad \forall \mathbf{x}_{u_1}, \ldots, \mathbf{x}_{u_j}, \quad \mathcal{P}(\mathbf{x}_1, \ldots, \mathbf{x}_n)}.$$

   **Two Predicates**. Let $\mathcal{P}(\mathbf{x}_1, \ldots, \mathbf{x}_n)$ and $\mathcal{Q}(\mathbf{y}_1, \ldots, \mathbf{y}_k)$ be predicates. Let $S \subseteq \{1, \ldots, n\} \times \{1, \ldots, k\}$ be a one-to-one sharing map, so that whenever $(i, j) \in S$, we identify the variables $\mathbf{x}_i$ and $\mathbf{y}_j$ by substituting $\mathbf{y}_j$ with $\mathbf{x}_i$.

   Define the merged variable set $\mathbf{M} := \{\mathbf{m}_1, \ldots, \mathbf{m}_{n+k-|S|}\}$ where the first $n$ variables are $\mathbf{x}_1, \ldots, \mathbf{x}_n$ in order, and the next $k - |S|$ variables are $\mathbf{y}_i$ variables where $i$ does not appear in the second component of any pair in $S$, indexed in ascending order. Let $\mathbf{m}_{\tau(i)}$ denote the variable in $\mathbf{M}$ corresponding to $\mathbf{y}_i$.

   Define the universal quantifier set $U \subseteq \{1, \ldots, n + k - |S|\}$ to be the set of indices of variables in $\mathbf{M}$ to quantify over. Let $\bar{U}$ denote the remaining indices of the free variables. Letting $U = \{u_1, \ldots, u_j\}$ and $\bar{U} = \{\bar{u}_1, \ldots, \bar{u}_{n+k-|S|-j}\}$, the production rule outputs a new predicate $\mathcal{R}$ such that

$$\texttt{forall } \mathcal{P} \, \mathcal{Q} \, S \, U \rightarrow$$

$$\boxed{\mathcal{R}(\mathbf{m}_{\bar{u}_1}, \ldots, \mathbf{m}_{\bar{u}_{|\bar{U}|}}) := \forall \mathbf{m}_{u_1}, \ldots, \mathbf{m}_{u_j}, \, \mathcal{P}(\mathbf{m}_1, \ldots, \mathbf{m}_n) \implies \mathcal{Q}(\mathbf{m}_{\tau(1)}, \ldots, \mathbf{m}_{\tau(k)}).}$$

5. **Match**: This rule allows variables to be set equal to each other. Let $\mathcal{A}(\mathbf{x}_1, \ldots, \mathbf{x}_n)$ be a function (resp. predicate), and let $I := \{i_1, \ldots, i_k\}$ with $i_1 < \ldots i_k$ be a set of indices to be matched. Let $J := (\{1, \ldots, n\} \setminus I) \cup \{i_1\} = \{j_1, \ldots, j_{n-k+1}\}$ with $j_1 < \ldots j_{n-k+1}$. Then the rule outputs a new function (resp. predicate) $\mathcal{B}$ with $n - k + 1$ arguments satisfying

$$\texttt{match } \mathcal{A} \, I \rightarrow \boxed{\mathcal{B}(\mathbf{x}_{j_1}, \ldots, \mathbf{x}_{j_{n-k+1}}) := \mathcal{A}(\mathbf{x}_1, \ldots, \mathbf{x}_n) \Big|_{\mathbf{x}_{i_2}, \ldots, \mathbf{x}_{i_k} \leftarrow \mathbf{x}_{i_1}}}$$

—that is, every occurrence of $\mathbf{x}_{i_2}, \ldots, \mathbf{x}_{i_k}$ in $\mathcal{A}$ is replaced by the variable $\mathbf{x}_{i_1}$.

6. **Constant**: This rule can turn an example into a concept (that accepts no inputs, i.e. a value concept). In FERMAT, we make the distinction between concepts and examples, and this rule provides a convenient way to bridge the gap. Let $e \in \mathcal{X}^+(m)$ be an example of the concept $m$. Then the production rule synthesizes a value concept $E$ out of this example,

$$\texttt{constant } e \rightarrow \boxed{E}$$

7. **Specialize**: This rule allows for specializing the input (for functions and predicates) or output (for functions).

   **Specialize Input.** Let $\mathcal{A}(\mathbf{x}_1, \ldots, \mathbf{x}_n)$ be a function (resp. predicate), and let $i$ be the index to specialize, and $v$ be the value to substitute. Then the rule outputs a new function (resp. predicate) $\mathcal{B}$ satisfying

$$\texttt{specialize } \mathcal{A} \ i \ v \rightarrow \boxed{\mathcal{B}(\mathbf{x}_1, \ldots, \mathbf{x}_{i-1}, \mathbf{x}_{i+1}, \ldots, \mathbf{x}_n) := \mathcal{A}(\mathbf{x}_1, \ldots, \mathbf{x}_{i-1}, v, \mathbf{x}_{i+1}, \ldots, \mathbf{x}_n)}$$

   **Specialize Output.** Let $\mathcal{F}(\mathbf{x}_1, \ldots, \mathbf{x}_n)$ be a function and let $v$ be the value we want to specialize the concept to. Then the rule outputs a new predicate $\mathcal{P}$ satisfying

$$\texttt{specialize } \mathcal{F} \ v \rightarrow \boxed{\mathcal{P}(\mathbf{x}_1, \ldots, \mathbf{x}_n) := (\mathcal{F}(\mathbf{x}_1, \ldots, \mathbf{x}_n) = v)}$$

8. **Negate**: Let $\mathcal{P}$ be a predicate, then this production rule outputs the negation of the predicate, i.e.

$$\texttt{negate } \mathcal{P} \rightarrow \boxed{\mathcal{Q} := \mathrm{Not}(\mathcal{P})}$$

9. **Size**: Let $\mathcal{P}(\mathbf{x}_1, \ldots, \mathbf{x}_n)$ be a predicate and $I = \{i_1, \ldots, i_m\}$ be a set of indices. Let $J := \{1, \ldots, n\} \setminus I = \{j_1, \ldots, j_{n-m}\}$ with $j_1 < \ldots j_{n-k}$. Then the production rule outputs a new concept $\mathcal{Q}$

$$\texttt{size } \mathcal{P} \ I \rightarrow \boxed{\mathcal{Q}(\mathbf{x}_{j_1}, \ldots, \mathbf{x}_{j_{n-m}}) := \#\{(\mathbf{x}_{i_1}, \ldots, \mathbf{x}_{i_m}) \,|\, \mathcal{P}(\mathbf{x}_1, \ldots, \mathbf{x}_n)\}}$$

   where $\#X$ denotes the cardinality of the set $X$.

**Conjecture Production Rules.**

1. **Implication**: Let $\mathcal{P}$ and $\mathcal{Q}$ be predicates over the same domain, each with $n$ inputs. Then the production rule outputs a conjecture

$$\texttt{implies } \mathcal{P} \ \mathcal{Q} \rightarrow \boxed{\forall \mathbf{x}_1, \ldots, \mathbf{x}_n, \mathcal{P}(\mathbf{x}_1, \ldots, \mathbf{x}_n) \implies \mathcal{Q}(\mathbf{x}_1, \ldots, \mathbf{x}_n)}$$

2. **Equivalence**: This rule conjectures equivalence of concepts.

   **Predicate.** Let $\mathcal{P}$ and $\mathcal{Q}$ be predicates over the same domain, each with $n$ inputs. Then the production rule outputs a conjecture

$$\texttt{equivalence } \mathcal{P} \ \mathcal{Q} \rightarrow \boxed{\forall \mathbf{x}_1, \ldots, \mathbf{x}_n, \mathcal{P}(\mathbf{x}_1, \ldots, \mathbf{x}_n) \iff \mathcal{Q}(\mathbf{x}_1, \ldots, \mathbf{x}_n)}$$

   **Function.** Let $\mathcal{F}$ and $\mathcal{G}$ be functions over the same domain, each with $n$ inputs. Then the production rule outputs a conjecture

$$\texttt{equivalence } \mathcal{F} \ \mathcal{G} \rightarrow \boxed{\forall \mathbf{x}_1, \ldots, \mathbf{x}_n, \mathcal{F}(\mathbf{x}_1, \ldots, \mathbf{x}_n) = \mathcal{G}(\mathbf{x}_1, \ldots, \mathbf{x}_n)}$$

3. **Nonexistence**: This rule asserts non-existence conjectures.

   **Predicate.** Let $\mathcal{P}(\mathbf{x}_1, \ldots, \mathbf{x}_n)$ be a predicate. Then the production rule outputs a conjecture

$$\texttt{nonexistence } \mathcal{P} \rightarrow \boxed{\nexists \mathbf{x}_1, \ldots, \mathbf{x}_n, \ \mathcal{P}(\mathbf{x}_1, \ldots, \mathbf{x}_n)}$$

   **Function.** Let $\mathcal{F}(\mathbf{x}_1, \ldots, \mathbf{x}_n)$ be a function and $v$ be a value. Then the production rule outputs a conjecture

$$\texttt{nonexistence } \mathcal{F} \ v \rightarrow \boxed{\nexists \mathbf{x}_1, \ldots, \mathbf{x}_n, \ \mathcal{F}(\mathbf{x}_1, \ldots, \mathbf{x}_n) = v}$$

4. **Exclusivity**: This production rule outputs conjectures stating that certain concepts are satisfied only on a particular finite set of inputs.

   **Predicate.** Let $\mathcal{P}(\mathbf{x}_1, \ldots, \mathbf{x}_n)$ be a predicate, and $S$ be a subset of $\mathrm{Domain}(\mathcal{P})$. Then the production rule outputs a conjecture

$$\texttt{exclusivity } \mathcal{P}\ S \rightarrow \boxed{\forall \mathbf{x}_1, \ldots, \mathbf{x}_n,\ \mathcal{P}(\mathbf{x}_1, \ldots, \mathbf{x}_n) \implies (\mathbf{x}_1, \ldots, \mathbf{x}_n) \in S}$$

   **Function.** Let $\mathcal{F}(\mathbf{x}_1, \ldots, \mathbf{x}_n)$ be a function and $v$ be a value. Then the production rule outputs a conjecture

$$\texttt{exclusivity } \mathcal{F}\ S\ v \rightarrow \boxed{\forall \mathbf{x}_1, \ldots, \mathbf{x}_n,\ \mathcal{F}(\mathbf{x}_1, \ldots, \mathbf{x}_n) = v \implies (\mathbf{x}_1, \ldots, \mathbf{x}_n) \in S}$$

A production rule application will propagate the input entities' computational implementation and examples where possible. While the produced symbolic definitions and computational implementations are deterministic, some rules have nondeterminism in the manner which new examples are adding upon creating.

## A.2 FERMAT Technical Details.

Here we describe further implementation details regarding FERMAT.

1. **Forbidden paths**: Following HR, we institute some forbidden paths which disallow the application of certain rules automatically. In particular, this disallows the creation of the following definitions & conjectures on input definition $P$: $[\neg\neg P, P \implies P, P \iff P, P \iff \neg P]$. Though this forms a minor optimization for our experiments, we believe that this set of forbidden paths can be learned automatically. In particular, in an extension of FERMAT which also allows interpretable proofs, an agent may quickly recognize these paths lead to uninteresting proofs and prevent their usage in the future.

2. **Global Instance Storage**: By instance of a theory, we refer to all concrete values of the domain introduced thus far in the theory. Our environment keeps track of all instances seen in the theory throughout the theory exploration process.

3. **Z3 Example Addition**: Definitions created involving the universal or existential quantifier rules cannot add examples or non-examples respectively. This is because adding such instances to the data of the entity requires iterating over all values of the Nat type, which is infinite. However, the more data an entity has the richer the theory. For such cases, we add certified examples for such definitions, by randomly sampling an instance of values, and using Z3 to determine whether it forms an example or non-example. We find this helps to prevent future nonexistence and trivial implication conjectures.

## A.3 Proving through Z3.

Our DSL allows users to define functions and predicates over bounded and unbounded parameters, and to compose them into logical conjectures using constructs such as `ForAll`, `Exists`, `Implies`, `And`, and arithmetic expressions. Critically, the DSL supports *nested definitions*: a predicate can define helper functions and other predicates inside its body, and similarly for functions. This enables the modular construction of conjectures and facilitates reuse of previously discovered building blocks.

**Compiler and SMT-LIB Translation.** The DSL is compiled to SMT-LIB, the input language accepted by Z3. Our compiler performs:

1. Flattening of nested definitions into top-level SMT functions,

2. Lexical scoping resolution and name hygiene to avoid collisions,

3. Translation of DSL-level constructs into logically equivalent SMT forms. We write a compiler which converts our DSL into the SMT-lib target language. The compiler is written using the `parglare` [14] library.

```
      f_0 := Func(
        params 1;
        bounded params 0;
        ReturnExpr 2 * x_0;
        ReturnPred None;
      );
      f_1 := Func(
        params 0;
        bounded params 0;
        ReturnExpr 6;
        ReturnPred None;
      );
      ReturnExpr None;
      ReturnPred Exists(
          [b_0],
          f_0(x_0=b_0) == f_1()
      );
```

(a) A DSL program asserting that 6 is even using function composition.

```
      p_0 := Pred(
        params 1;
        bounded params 1;

        f_0 := Func(
          params 1;
          bounded params 0;
          ReturnExpr x_0 + 1;
          ReturnPred None;
        );

        ReturnExpr None;
        ReturnPred Exists(
          [b_0],
          f_0(x_0=b_0) == x_0
        );
      );
```

(b) A nested DSL predicate defining an inner function and using it in an existential condition.

Figure 5: DSL snippets

```
(define-fun f_0_p_0 ((x_0 Int)) Int (+ x_0 1))
(define-fun p_0 ((x_0 Int)) Bool
  (exists ((b_0 Int)) (= (f_0_p_0 b_0) x_0)))
```

Figure 6: SMT-LIB code generated from Figure 5b, where naming collisions are avoided by hygienic flattening.

**Semantics and Proof Feedback.** When a conjecture is compiled and passed to Z3, the prover returns one of three outcomes:

UNSAT — The conjecture is logically valid. It is added to the theory as a verified theorem and can be used in future derivations.

SAT — The conjecture is invalid. Z3 returns a counterexample, which is parsed back into the DSL domain.

Timeout — We used 2s timeout with the Z3 solver.

**Theory-Guided Composition.** The DSL serves as a unifying language in our system: all definitions, lemmas, and conjectures are expressed as DSL programs. As new concepts are discovered by our theory formation framework (FERMAT), they are registered as definitions in the DSL. New conjectures are then automatically generated by composing these building blocks. This compositional ability—enabled by nesting and a hygienic compiler—allows our system to express and verify arbitrarily structured mathematical ideas.

## A.4   Ground Truth Set

We curated 180 ground truth functions, theorems, and conjectures using a well-known introductory elementary number theory text [2] as well as a small set of famous conjectures in the number theory literature, and an additional 67 ground truth entities drawn from the theory of finite fields over $\mathbb{F}_{27}$. These ground truth concepts can be entirely derived by applying the production rules to the base concepts ( zero, successor, and equality for number theory, and generators and field operations for $\mathbb{F}_{27}$). Figure 7 illustrates a small subset of ground truth concepts that relate to divisibility, demonstrating an area in which the model would receive extrinsic reward for discovering the most basic properties of natural numbers. On the other hand, Figure 8 illustrates a small subset of ground truth concepts relating to more sophisticated theorems and conjectures in the theory of prime numbers which offer extrinsic reward in theorizing about abstract ideas such as the existence of infinite instances. Figure 9 covers a subset of definitions and theorems in $\mathbb{F}_{27}$.

It is worth noting that there may be multiple paths to arriving at a single concept. For example, it is possible to derive the concept is even either by applying:

```
apply specialized divides two index_to_specialize=0
```

which gives a new concept with the first argument of the divides function to two, or by applying:

```
apply exists double indices_to_quantify=0
```

which gives a new concept that returns True for an input if there exists a natural number such that doubling it results in the input. The number of possible paths to reach a certain ground truth concept increases exponentially with complexity. Because we want to evaluate the algorithm on its ability to find a ground truth concept regardless of the path it takes, we included redundant concepts in our ground truth set. In this manner, we cover as many paths to meaningful math concepts as possible, and we provide a smooth reward signal to the algorithm for defining interestingness.

## A.5   Computational Resources & Hyperparameters

Our experiments are run on 64 Intel Xeon Platinum 8352Y and 64 AMD EPYC 7413 24C CPUs. We leverage parallelism built into FERMAT to enable speedup in the evaluations. Given this allocation, evaluating a single interestingness measure through episodic rollouts with FERMAT using the configuration detailed in  takes $\sim M = 120$ seconds when using 64 workers. Our FunSearch and EvoAbstract experiments take significantly longer due to large number of interestingness measures generated and evaluated during evolutionary search. In particular, each experimental result reported with either FunSearch/EvoAbstract takes approximately 18 hours with 64 workers. The evaluation of our GPT-4o baseline takes a total of 6 hours when using 64 workers.

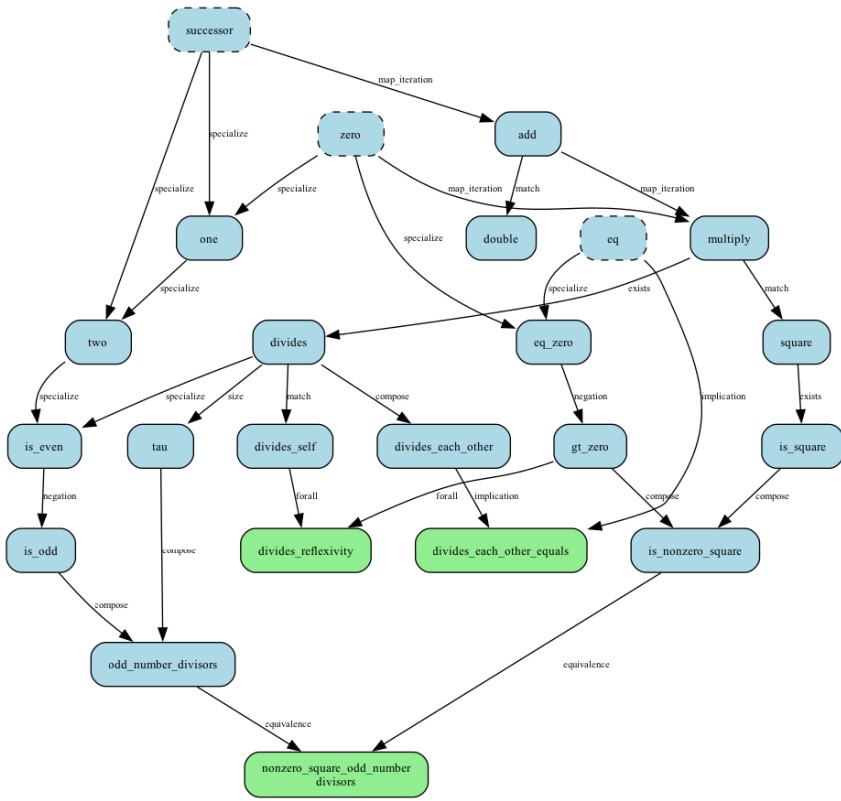

Figure 7: Sample knowledge graph of ground truth entities relating to basic properties of divisibility in the domain of natural numbers.

---

**Algorithm 1** EvoAbstract: Synthesis via Evolution and Abstraction Learning

---

**Require:** Template $T$; Number of islands $k$; Generations $N_{gen}$; Abstraction frequency $G \in \mathbb{N}^+$;
**Require:** Parent sample size $n_p \in \mathbb{N}^+$; Abstraction candidate sample size $n_{abs} \in \mathbb{N}^+$; Evolution sample size $n_e$;
**Require:** Evolution & Abstraction LLMs $\mathcal{L}_{var}, \mathcal{L}_{abs}$;
 1: Initialize $k$ populations $\mathcal{P}_1, \ldots, \mathcal{P}_k$ with seed programs.
 2: Initialize $k$ empty abstraction libraries $\text{Lib}_1, \ldots, \text{Lib}_k$.
 3: **for** generation $g = 1$ to $N_{gen}$ **do**
 4:     Sample island $i \sim \text{Uniform}\{1, \ldots, k\}$.
 5:     $P \leftarrow \text{EVOLUTIONSTEP}(\mathcal{P}_i, T, \text{Lib}_i, n_p, n_e, \mathcal{L}_{var}, \text{Scores})$.
 6:     $\text{Score} \leftarrow \text{POLICYEVALUATIONSTEP}(P)$.
 7:     Update population $\mathcal{P}_i \leftarrow \mathcal{P}_i \cup \{P\}$
 8:     **if** $g \bmod G \equiv 0$ **then**                    ▷ Perform abstraction phase periodically
 9:         **for all** islands $i = 1$ to $k$ **do**
10:             $\text{Lib}_i \leftarrow \text{Lib}_i \cup \text{ABSTRACTIONSTEP}(\mathcal{P}_i, T, \text{Lib}_i, n_{abs}, \mathcal{L}_{abs}, \text{Scores})$.
11: **return** Best program $f^*$ found across all populations $\mathcal{P}_1, \ldots, \mathcal{P}_k$.

---

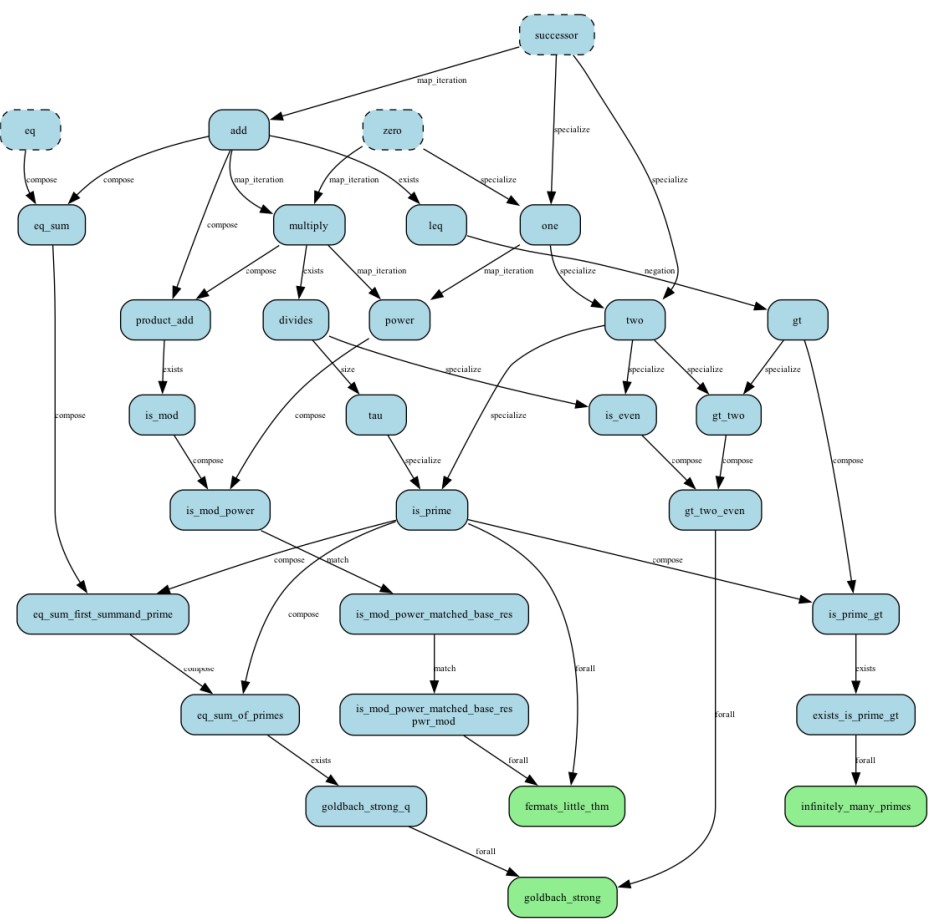

Figure 8: Sample knowledge graph of ground truth entities relating to theorems and conjectures central to the theory of prime numbers. Note that the concept of primality, `is_prime`, is an ancestor of many concepts.

---

**Algorithm 2** Policy Template for Action Selection

---

**Require:** Interestingness measure $\mathcal{I}(\text{entity}, \text{graph}) \mapsto \mathbb{R}$.
**Require:** Knowledge Graph $G = (V, E)$ with definitions $\mathcal{D} \subset V$.
**Require:** Number of definitions to sample $N \in \mathbb{N}^+$.
**Require:** Simulation limit $S_{lim} \in \mathbb{N}^+$.
1: $\mathcal{D}_{sampled} \leftarrow \emptyset$          ▷ Set of $N$ sampled definitions
2: $Scores \leftarrow \{\}$          ▷ Map each definition to its interestingness score
3: **for** each definition $d \in \mathcal{D}$ **do**
4:      $Scores[d] \leftarrow \mathcal{I}(d, G)$
5: **end for**
6: $\mathcal{D}_{sampled} \leftarrow \text{SampleByScore}(Scores, N)$
7: $\mathcal{A}_{potential} \leftarrow \text{EnumeratePossibleActions}(\mathcal{D}_{sampled}, G)$
8: $\mathcal{A}_{sim} \leftarrow \text{Sample}(\mathcal{A}_{potential}, \min(S_{lim}, |\mathcal{A}_{potential}|))$    ▷ Randomly sample up to $S_{lim}$ actions
     for simulation
9: $SimulatedActionScores \leftarrow \{\}$
10: **for** each action $a \in \mathcal{A}_{sim}$ **do**
11:      $e_{new} \leftarrow \text{SimulateAction}(a, G)$       ▷ Simulate action $a$ to get resulting entity $e_{new}$
12:      $score_a \leftarrow \mathcal{I}(e_{new}, G)$          ▷ Compute interestingness of the new entity
13:      $SimulatedActionScores[a] \leftarrow score_a$
14: **end for**
15: $a^* \leftarrow \text{SampleByScore}(\mathcal{A}_{sim}, 1)$
16: **return** $a^*$

---

Figure 9: Sample knowledge graph of ground truth entities relating to theorems and conjectures central to the theory of the finite field $\mathbb{F}_{27}$.

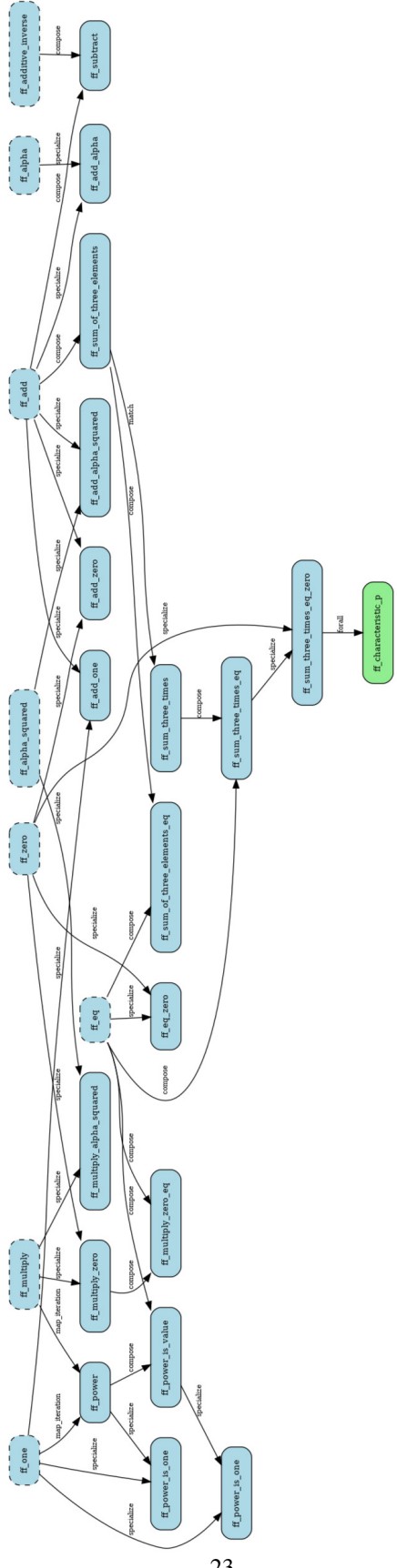

Figure 10: The one-shot prompt for our GPT-4o baseline. We do not insert the primitives here for brevity, these can be found in Figure 11.

Figure 11: A list of primitive methods available to the interestingness measure synthesizers. Each method returns a simple property or information about the knowledge graph and/or the input entity.

We note that our episodic roll-outs are *time-capped*, not finishing after a limit number of steps. This is because there is a natural variance in the types of mathematical entities that get constructed, and calls to the Z3 theorem prover can often take many seconds. When resolving conjectures, we set the timeout to Z3 to be $2.0$ seconds, and to $0.5$ seconds when using Z3 to add instances to entities without any.

## A.6 REPL

FERMAT also comes equipped with a read-eval-print-loop (REPL) for manual interaction with the environment. The REPL allows the user to use an interactive shell to define, inspect, and evaluate mathematical entities.

**Available commands.**

| Command | Description |
|---------|-------------|
| help | Get help on commands or list available rules |
| list | List available concepts, rules, or conjectures |
| apply | Apply a production rule to create new concepts/conjectures |
| inspect | Show detailed information about an entity |
| compute | Test computational implementation with arguments |
| rename | Rename an entity |
| remove | Remove an entity |
| visualize | Create a visualization of the current knowledge graph |
| clear | Clear the screen |
| save | Save knowledge graph to file |
| exit | Exit the REPL |

**Example usage.**

| REPL Command | Concept Produced |
|--------------|------------------|
| apply iter successor | *add* |
| apply iter add zero | *multiply* |
| apply match add indices_to_match=[0,1] | *double* |
| apply match multiply indices_to_match=[0,1] | *square* |
| apply specialize successor zero index_to_specialize=0 | *one* |

Figure 12: The prompt supplied to the evolution sampler $\mathcal{L}_{var}$, indicating the evolution task that needs to be applied. We have removed the description of the DSL primitives which appears in 11.

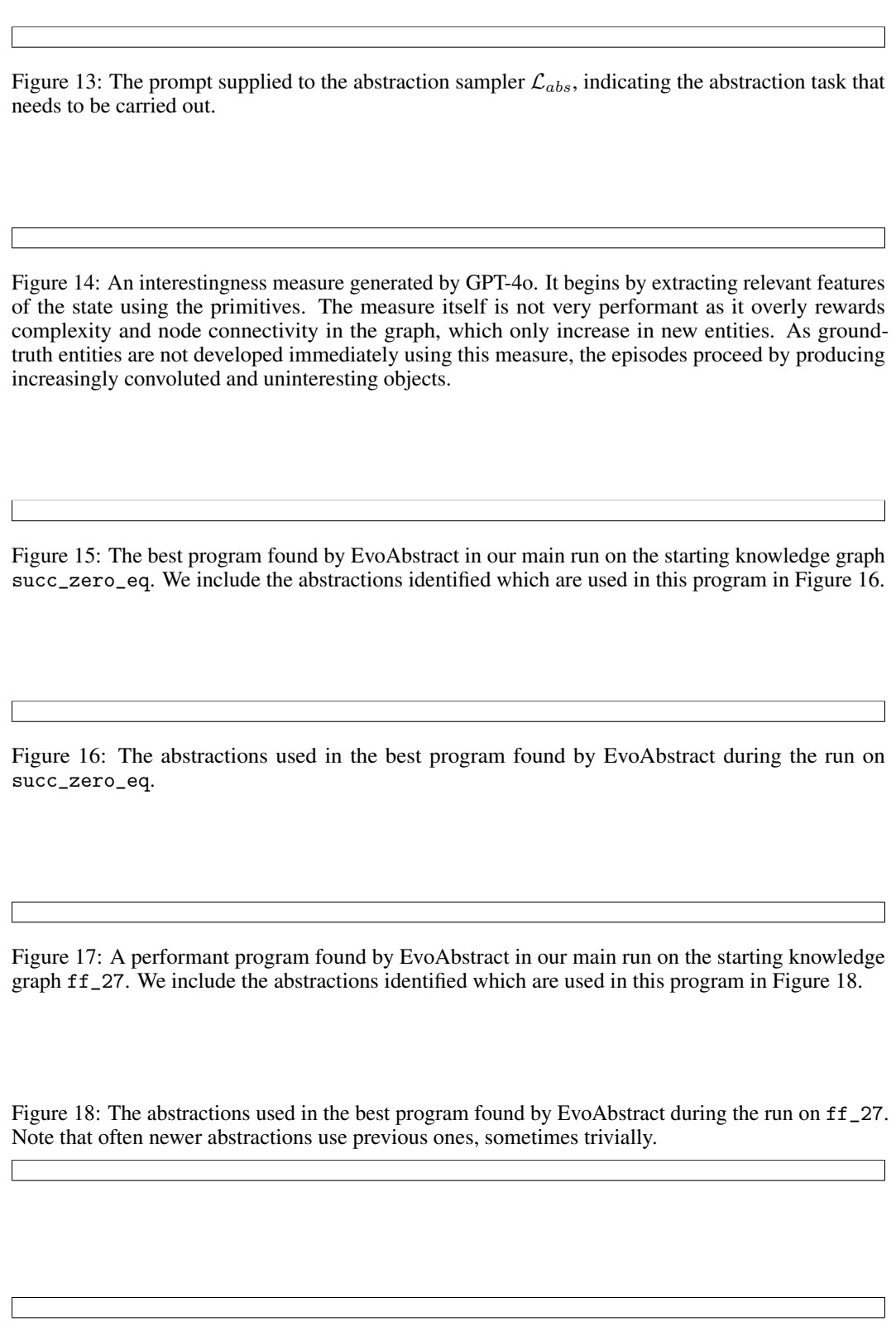

Figure 13: The prompt supplied to the abstraction sampler $\mathcal{L}_{abs}$, indicating the abstraction task that needs to be carried out.

Figure 14: An interestingness measure generated by GPT-4o. It begins by extracting relevant features of the state using the primitives. The measure itself is not very performant as it overly rewards complexity and node connectivity in the graph, which only increase in new entities. As ground-truth entities are not developed immediately using this measure, the episodes proceed by producing increasingly convoluted and uninteresting objects.

Figure 15: The best program found by EvoAbstract in our main run on the starting knowledge graph `succ_zero_eq`. We include the abstractions identified which are used in this program in Figure 16.

Figure 16: The abstractions used in the best program found by EvoAbstract during the run on `succ_zero_eq`.

Figure 17: A performant program found by EvoAbstract in our main run on the starting knowledge graph `ff_27`. We include the abstractions identified which are used in this program in Figure 18.

Figure 18: The abstractions used in the best program found by EvoAbstract during the run on `ff_27`. Note that often newer abstractions use previous ones, sometimes trivially.

Figure 19: The best program found by FunSearch during the run on `succ_zero_eq`.

Figure 20: Sample sections of elementary number theory discovered by EvoAbstract during runs on `succ_zero_eq` and `arith_base`.

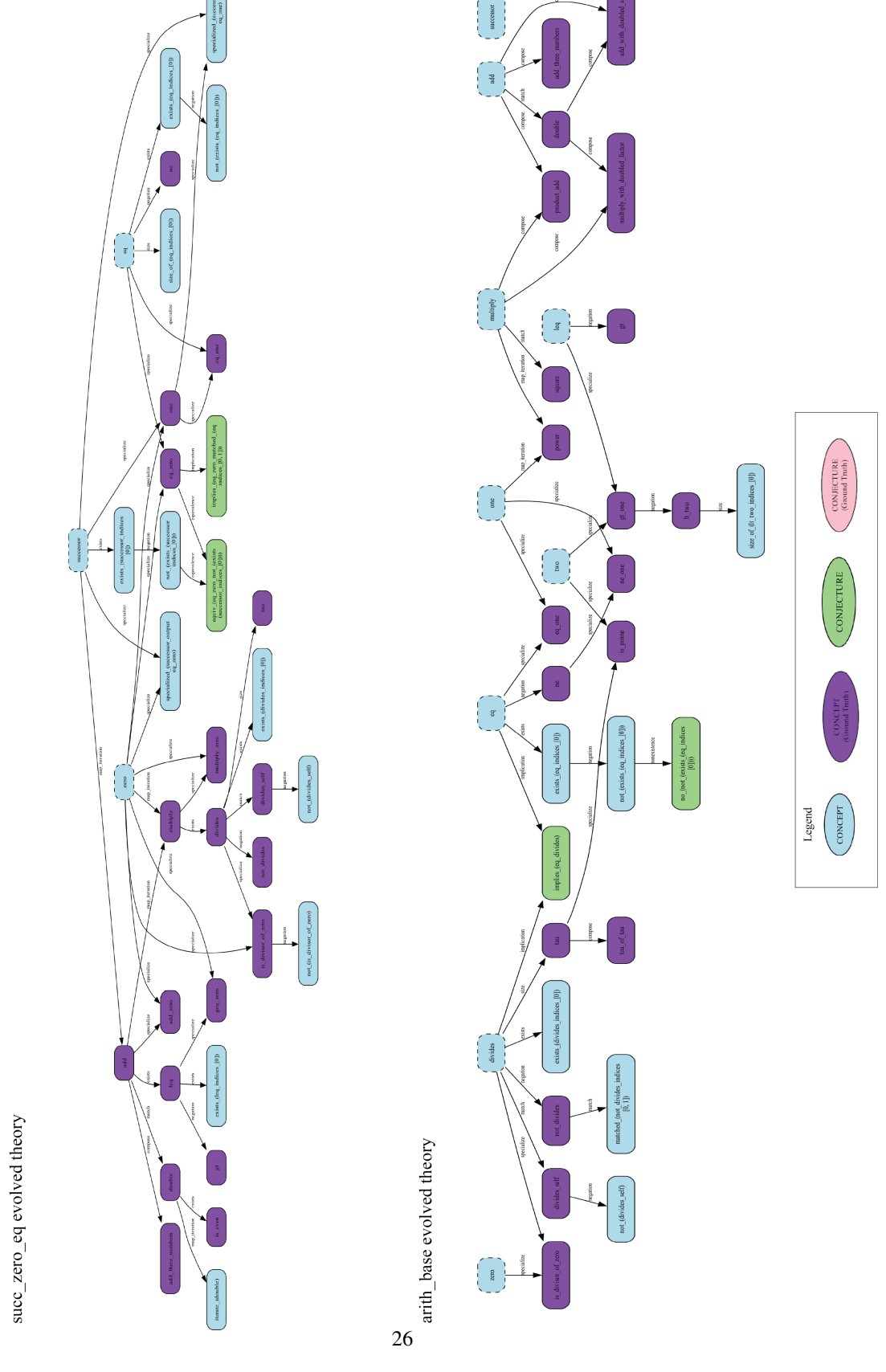

