# OpenReview forum: "Learning Interestingness in Automated Mathematical Theory Formation"
_NeurIPS.cc/2025/Conference — NeurIPS 2025 spotlight_

### Official Review · Reviewer_c2rm · 2025-07-01

**Clarity:** 2
**Significance:** 2
**Originality:** 3
**Rating:** 4
**Confidence:** 4

**Summary:**

This paper introduces FERMAT, a reinforcement learning (RL) framework for automated mathematical theory formation, and EvoAbstract, an LLM-driven evolutionary algorithm to learn "interestingness" measures for guiding theory discovery. FERMAT models theory formation as a Markov Decision Process (MDP), enabling systematic exploration of mathematical concepts, conjectures, and theorems. EvoAbstract synthesizes interestingness metrics by integrating function abstraction, outperforming hard-coded baselines (e.g., HR measures) in discovering 180 ground-truth number theory entities. Key contributions include: (1) formalizing theory formation as an MDP with symbolic actions; (2) an abstraction-learning mechanism for modular program synthesis; (3) empirical validation showing EvoAbstract discovers 20.25 (succ_zero_eq) and 39.74 (arithmetic_base) ground-truth entities, surpassing baselines.

**Questions:**

1. Can FERMAT/EvoAbstract be extended to algebra or analysis? A case study on ring theory concepts (e.g., ideals) would validate generalizability.
2. Are plans to automate production rule discovery (beyond fixed rules in Appendix A.1)? Demonstrating rule evolution would enhance autonomy .
3. How might integrating Lean’s dependent types improve entity representation? An ablation on type-aware vs. type-agnostic interestingness would clarify impact .
4. Can EvoAbstract use lightweight neural networks instead of GPT-4o? Reducing LLM dependency would improve accessibility .
5. How does FERMAT handle knowledge graph redundancy as theories expand? Metrics on graph efficiency (e.g., cycle detection) are needed .

**Ethical Concerns:**

["Major Concern: Data privacy, copyright, and consent"]

**Final Justification:**

After reviewing the rebuttal and author clarifications, I have decided to maintain my original score of 4. The paper introduces a compelling and original framework (FERMAT + EvoAbstract) for formalizing automated mathematical theory formation, and contributes to a relatively underexplored but high-potential research area. However, several concerns regarding evaluation scope, practical reproducibility, and clarity remain only partially resolved, and in my view, they currently outweigh the strengths of the submission. Below is a breakdown of my reasoning:

1.  The authors added a preliminary extension to Galois fields in the rebuttal, which is promising. While limited in scope, this demonstrates potential for generalization.
GPT-4o dependency and reproducibility
2. The authors clarified that cost was minimal and other LLMs (e.g., Qwen, Gemini) could serve as alternatives. This improves accessibility, though results may still vary across LLMs.
Lean integration and dependent types
3. Authors explained the extensibility of their DSL and acknowledged the benefits of incorporating Lean in future work. This is a reasonable response given the paper's current scope.

**Limitations:**

The authors acknowledge technical limitations (scalability, domain scope) but underaddress societal impacts. Suggestions:

(1). Discuss risks of misusing FERMAT to generate flawed mathematical theories in safety-critical applications (e.g., cryptography verification).
(2). Address biases in the curated ground-truth set, which may favor Western mathematical traditions over others.

**Paper Formatting Concerns:**

No major formatting issues. The paper adheres to NeurIPS guidelines, with clear equations, tables, and appendices .

**Quality:**

3

**Strengths And Weaknesses:**

**Strengths:**

1. FERMAT’s MDP formalization unifies theory formation steps (definition production, proof attempts) under a standard RL framework .
2. EvoAbstract’s abstraction learning (e.g., reusable subroutines like calculate_example_balance) improves modularity and performance by 50% over FunSearch .
3. Rigorous comparisons across baselines (Random, HR, GPT-4o) and ablation studies (FunSearch) validate component contributions .
Practical Impact: FERMAT’s REPL interface and open-source code enable reproducibility and real-world use in educational theory exploration .

**Weaknesses:**

1. FERMAT does not leverage dependent type systems (e.g., Lean), limiting expressiveness for complex mathematics .
2. EvoAbstract relies on GPT-4o, introducing bias and high computational cost for reproduction .
3. Evaluations focus on elementary number theory, lacking validation in algebra or analysis .

---

> ### Author Rebuttal · Authors · 2025-07-31
>
> Thank you for your review of our paper! We appreciate the recognition of our unification of theory formation into an MDP problem, and the rigorous comparisons of baselines, as well as the practical impact of our framework for enabling further research in this poorly studied area. We respond to your comments and questions below:
>
> > FERMAT does not leverage dependent type systems (e.g., Lean), limiting expressiveness for complex mathematics .
>
> **Response**: There is no fundamental limitation in Fermat that prevents the use of Lean. We have done the heavy lifting of supporting general definition composition through a custom generic DSL and compiler, which is a nontrivial task (see Appendix A.3 on page 16, lines 647-669). The DSL is designed in such a way that it can be cross-compiled to any target language such as Lean 4, Z3, Rocq. We chose to implement Z3 as one of the possible target languages because it comes with automated theorem-proving capabilities. In particular, Z3 serves as a black-box prover so as to reduce the number of moving pieces in our work, which is an exploration in automated theory formation that has not really been researched in the modern era. We find that our evolutionary method is useful for theory formation, but at present it seems that there are still many bottlenecks to scaling to more complex math, which we are working on and hope to expose to the community through this work.
>
> Furthermore, exploring more complex mathematics will require a more sophisticated prover, as automation like Z3 does not scale to more complex statements. Hence, learning to prove from scratch (with low data) will be necessary, and this in itself is another significant research direction which is an excellent source for future work.
>
> > EvoAbstract relies on GPT-4o, introducing bias and high computational cost for reproduction.
> > Can EvoAbstract use lightweight neural networks instead of GPT-4o? Reducing LLM dependency would improve accessibility .
>
> **Response**: Thanks for raising this! Our experiments are relatively inexpensive - reproducing all results in the paper required 2,000 samples, costing under 20 USD with GPT-4o-mini. Open-source or cheaper APIs like Qwen 3 Turbo (0.20 USD/million tokens) or free providers like Gemini make reproduction even more accessible. Any method that can generate Python programs from instructions would work - we use LLMs because they’re effective.
> Regarding bias, could you clarify what you meant? If it's data contamination, the LLM only sees past programs and their scores - never the domain or generated concepts. If you meant social bias in the LLM, our setup is narrowly scoped and we believe such effects are negligible.
>
> >  Evaluations focus on elementary number theory, lacking validation in algebra or analysis .
>
> **Response**: This is a great point! We agree that another domain would help illustrate the benefits of our method and provide another off-the-shelf exploration point for the community. We liked your suggestion of exploring algebra, and so to demonstrate the extensibility of the FERMAT framework for our rebuttal, we have implemented the theory of Galois fields. Like in elementary number theory, we have curated a list of 50 ground truth entities for this domain, to be included in the revision.
>
> We reran the baseline experiments (with lesser compute of 64 iterations with 30 seconds to an episode due to the time constraint) and confirmed that our evolutionary approach learns interestingness functions that outperform fixed HR measures.
> Among hardcoded metrics, comprehensibility performed best with a reward of 4.48, while equal weighting scored 3.00. Funsearch found a measure scoring 6.52, while our method, EvoAbstract, achieved 7.78, slightly ahead of Funsearch.
> We plan to include more experiments with this domain in our next revision, especially to measure the variance of evolution runs.
>
> > Are plans to automate production rule discovery (beyond fixed rules in Appendix A.1)? Demonstrating rule evolution would enhance autonomy .
>
> **Response**: Yes! This is a very promising research direction. For example a very common pattern in math is defining an object that satisfies an extremal property, such as integers with smallest number of non-unit divisors (i.e. primes), or elements of a finite field with the highest order (i.e. primitive elements). This is a common structure and the prospect of learning such production rules is promising! We think this is a good direction for future work. One good historical reference would be Eurisko (1983), which extended the AM system by trying to learn production rules automatically with some success for VLSI chip design and naval fleet simulations.
>
>
>
> > How might integrating Lean’s dependent types improve entity representation? An ablation on type-aware vs. type-agnostic interestingness would clarify impact .
>
> **Response**: Good point! Indeed having Lean's dependent types should allow for more expressive representation. We do make use of a simple type system through our DSL which informs the values of certain interestingness measures, like parsimony, which is a basic measure involving the arity of the entity. Though, in our experiments, the parsimony measure does not actually perform better than a random baseline, indicating that this particular type information may not provide sufficient signal for replicating the ground truth set. That being said, extensions to using Lean should be able to benefit from richer information about entities that should better inform measures of interestingness, which we think is a great subject for future work. Similarly the richer proving environment lends itself to interpretable proofs, which themselves can be mined for properties to better judge the interestingness of statements. As mentioned before, since we use a custom DSL that can be cross-compiled to any target language, we can easily add support for Lean. The more expressive types available in Lean will be useful for more fine-grained representations of entities.
>
> >How does FERMAT handle knowledge graph redundancy as theories expand? Metrics on graph efficiency (e.g., cycle detection) are needed.
>
> **Response**: Good question! By construction, the knowledge graph is acyclic. Each entity has a description length - the number of characters needed to express it using production rules starting from base definitions. Since applying a rule always increases this length, cycles (which would require the length to decrease) are impossible. This prevents syntactic redundancy.
> Semantic redundancy is harder to eliminate. Fermat removes one of two entities when it proves them equivalent, but this relies on the agent proposing such conjectures. Unlike older systems like HR, which exhaustively checked equivalence (at high cost), we avoid hard-coding and instead aim to learn this behavior.
>
>
>
> > (1). Discuss risks of misusing FERMAT to generate flawed mathematical theories in safety-critical applications (e.g., cryptography verification). (2). Address biases in the curated ground-truth set, which may favor Western mathematical traditions over others.
>
> Thank you for raising these important points. While FERMAT is not yet capable of generating novel mathematical theories, we recognize that automated theory formation raises significant societal and safety concerns. We believe that such systems must include formal guarantees and human oversight, especially in safety-critical settings where specification is essential. In this paper, our benchmark draws from widely taught concepts in elementary number theory, which form a global mathematical foundation with minimal cultural bias. We'll be sure to include this discussion on societal impact in our next revision.
>
> Thank you for your insightful review! We would be happy to answer any further questions you might have!
>
> [1] Eurisko: A Program That Learns New Heuristics and Domain Concepts, Douglas Lenat, 1983
> [2] HR: Automatic Concept Formation in Pure Mathematics, Simon Colton, Alan Bundy, Toby Walsh, 2000.

---

### Official Review · Reviewer_RkzN · 2025-07-02

**Clarity:** 2
**Significance:** 3
**Originality:** 3
**Rating:** 5
**Confidence:** 4

**Summary:**

This paper presents Fermat, a RL environment that formally models the formation of mathematical definitions, conjectures, and proofs of conjectures, and EvoAbstract, an evolutionary approach for finding intrinsic reward functions that encourage exploration in Fermat that leads to the proving of certain known definitions, conjectures, and theorems. Fermat works over states modeled through graphs that represent definitions, conjectures, and theorems, and policies update these states by taking a variety of actions (new definitions, conjectures, and proof attempts, backed up by an automated prover). EvoAbstract uses an evolutionary approach that includes defining new intrinsic reward metrics while learning and using an abstraction library. The paper presents experiments that start with two different initial theories, comparing performance with a range of hardcoded intrinsic reward and a prior evolutionary search algorithm.

**Questions:**

Covered in significance concerns above.

**Ethical Concerns:**

["NO or VERY MINOR ethics concerns only"]

**Final Justification:**

As indicated below, I'm revising my score up.

**Limitations:**

yes

**Quality:**

3

**Strengths And Weaknesses:**

Strengths

+Uniqueness: to my knowledge, this sort of “automated theory-building” with definition and conjecture formation, in addition to theorem proving, in which an interestingness score is learned, has not been done before.

+Significance: Mathematical theorem proving is of broad interest, as is conjecture formation. Recently, intrinsic reward has been used to guide conjecture formation. This work should hence be of interest to both the theorem proving and open-ended learning communities. The experimental results show considerable improvement over baseline approaches — in particular, the introduction of abstraction learning and seeing the performance boost over FunSearch is really helpful.

+Quality: the Fermat environment provides an interesting, interpretable environment for theory-building.

Weaknesses

-Clarity: while much of the paper was fairly clear, aspects took quite a bit of time to understand. Minor example: you describe the computational interpretation as being a function to evaluate whether an input satisfies a given symbolic definition, which makes me think that it should return a bool, but the example simply implements an operation, returning a number. More serious example: I found 2.2 to be really opaque. In particular, you describe factoring a policy space into two spaces, a space of policy templates and a space of intrinsic reward functions. It was really unclear, at that point, what the class of policy templates might be. My understanding is that you actually just have a simple fixed algorithm for getting a policy from an intrinsic reward (sampling and simulating, using the interestingness of the outcome to choose, Alg 2), but I could only understand this by being directed to Alg 2 which is referenced in section 5.

-Significance: the experimental comparisons, aside from that with FunSearch, seem like a foregone conclusion. These all have hardcoded intrinsic reward functions, whereas FunSearch and EvoAbstract get to optimize over generations. This might be surprising if the intrinsic reward functions were evolved from scratch, but they get to lean on GPT-4o, and the GPT-4o baseline demonstrates that this is a fairly strong starting point. It would be interesting to see how well evolutionary search works if it has a weaker starting-point, or if evolving intrinsic reward is good for theorem-proving (as opposed to directly evolving policies, if possible), or if other forms of open-ended learning (e.g. adapting Minimo to a fine-tuned LLM setting) might be better.

I think it’d also be helpful to relate this work to others on evolving intrinsic motivation [1, 2] and meta-RL [3].

[1] Singh, Satinder, Richard L. Lewis, Andrew G. Barto, and Jonathan Sorg. "Intrinsically motivated reinforcement learning: An evolutionary perspective." *IEEE Transactions on Autonomous Mental Development* 2, no. 2 (2010): 70-82.

[2] Alet, Ferran, Martin F. Schneider, Tomas Lozano-Perez, and Leslie Pack Kaelbling. "Meta-learning curiosity algorithms." *arXiv preprint arXiv:2003.05325* (2020).

[3] Norman, Ben, and Jeff Clune. "First-Explore, then Exploit: Meta-Learning to Solve Hard Exploration-Exploitation Trade-Offs." *Advances in Neural Information Processing Systems* 37 (2024): 27490-27528.

---

> ### Author Rebuttal · Authors · 2025-07-31
>
> Thank you for your thorough review! We appreciate your comments on the novelty of our research problem, and the significance of our effort to study it and produce the open-source framework that should enable further research into this area! We respond to your comments and questions below:
>
> > I found 2.2 to be really opaque. In particular, you describe factoring a policy space into two spaces, a space of policy templates and a space of intrinsic reward functions. It was really unclear, at that point, what the class of policy templates might be. My understanding is that you actually just have a simple fixed algorithm for getting a policy from an intrinsic reward (sampling and simulating, using the interestingness of the outcome to choose, Alg 2), but I could only understand this by being directed to Alg 2 which is referenced in section 5.
>
> **Response**: Thank you for pointing this out. Indeed, the motivation for factoring the space into two components was to better separate the problem of measuring the value of a single entity and then scaffolding the intrinsic measure into a policy to be followed. This is generally consistent with prior work (like HR (2000), AM (1977)), and with our objective of learning how to measure the interestingness of entities. Fixing a policy template, we focus on automatically synthesizing this intrinsic reward measure. We agree with you that 2.2 as written is confusing. We plan to include an example of the purpose of this separation in S 2.2 to clarify it for the reader and make the overall reading flow more consistently. We also think it would be good to motivate the framing of previous work like HR and AM before reaching section 2.2, but we found this hard to do fully due to the limited page constraint. Happy to hear any thoughts you might have on this!
>
> > the experimental comparisons, aside from that with FunSearch, seem like a foregone conclusion. These all have hardcoded intrinsic reward functions, whereas FunSearch and EvoAbstract get to optimize over generations. This might be surprising if the intrinsic reward functions were evolved from scratch, but they get to lean on GPT-4o, and the GPT-4o baseline demonstrates that this is a fairly strong starting point. It would be interesting to see how well evolutionary search works if it has a weaker starting-point, or if evolving intrinsic reward is good for theorem-proving (as opposed to directly evolving policies, if possible), or if other forms of open-ended learning (e.g. adapting Minimo to a fine-tuned LLM setting) might be better.
>
> **Response**: This is a fair point, though we want to mention that it is somewhat unclear a priori whether evolutionary methods *can* optimize intrinsic reward measures in this setting. We use the LLM in a rather indirect way, only showing it previous programs produced, and the numerical score corresponding to the extrinsic reward. It never sees any concepts or examples of theories produced, and does not even know that it is working on elementary number theory. The reward signal is not necessarily very clear, yet the evolutionary process is still capable of optimizing it. We think this is a valuable finding.
>
> Indeed the suggestions you make regarding intrinsic reward for theorem-proving and targeting other forms of open-ended learning are interesting research avenues. These ideas should yield very fruitful research, though may be out of scope for our singular submission. We mention that we view our research contributions as: showing that the problem of theory-formation can be studied in the modern era with modern perspectives, that evolutionary methods are valuable for assessing interestingness (though not necessarily optimal!), and provide a framework to encourage broader community involvement in this problem. Of course, future research directions we included in the conclusion and that you are mentioned are topics we intend on exploring.
>
> In principle, the evolution operator only requires the ability to produce Python programs. Could you suggest some specific weaker operators that we could use for ablations? We chose GPT-4o-mini in our experiments to tradeoff cost (which we want to minimize) and ability to adhere to formatting instructions (which we want to maximize). LLMs happen to be a good option for this task, and we use them in a way that is indirect to the task.
>
> >  I think it’d also be helpful to relate this work to others on evolving intrinsic motivation [1, 2] and meta-RL [3].
>
> **Response**: Thank you for these excellent references! We will be sure to include them in our next revision along with a full discussion of their relation to our work. It is very encouraging to see AM (1977) as one of the central AI works motivating [1]. We find that their framework for an agent that interacts with its internal environment, then updated by interactions with the external environment, forms a general setting for our work which we do for the mathematical setting. In their case, the assumption with the optimal reward function that they seek to find is similar in our case. We make the Platonist assumption that such an optimal reward function exists (for simpler mathematics, at least) and can accurately classify all the ground truth entities in our benchmark. Indeed we find there is room in our experiments for improvement, as they also note that fitness-based reward functions still maintain a gap to the best reward. [2] is also relevant to our work in that they learn curiosity mechanisms that have some symbolic components through compositions and combinations of neural modules with buffers & other loss functions.
>
> We find the strategy in [3] to be very sensible. While the experiments done with Fermat in our submission do not explicitly attempt to have an exploratory stage to begin with, we believe that implicitly this sort of process is occurring. Initial rounds in the evolutionary procedure are marked by the model not knowing how to optimize on the reward, so the programs generated are usually quite poor, and can be thought of as some exploration. Once some high-performing programs are found, the LLM then has better signal on how to improve the reward. However, we don´t expect that the evolutionary operator is knowingly being exploratory. We think explicitly imposing this structure on the evolutionary process could be very useful for quicker convergence and better overall learning. Thank you again for suggesting these works, the context they provide for our work is excellent!
>
> > Minor example: you describe the computational interpretation as being a function to evaluate whether an input satisfies a given symbolic definition, which makes me think that it should return a bool, but the example simply implements an operation, returning a number
>
> Thank you for this observation, we agree that the description of the computational interpretation is a bit confusing, we will revise this by indicating the computational interpretation can evaluate the truth value of a predicate for certain inputs and also compute the values of operators in a subsequent revision.
>
> Thank you again for your detailed review of our paper, we really appreciate it! Happy to discuss/respond to any further questions!

---

> > ### Comment · Reviewer_RkzN · 2025-08-01
> >
> > Thanks much for your replies!
> >
> > Thanks for acknowledging the points about clarity -- it sounds like you understand well how to clarify. I do think it sounds like it'd be helpful to talk about how HR and AM lead up to this point, if only for a couple of sentences. I think that making this point really clearly will help give an immediate sense that the methods really get at testing whether evo algorithms can optimize intrinsic reward here.
> >
> > > We mention that we view our research contributions as: showing that the problem of theory-formation can be studied in the modern era with modern perspectives, that evolutionary methods are valuable for assessing interestingness (though not necessarily optimal!), and provide a framework to encourage broader community involvement in this problem.
> >
> > The comments you made here are helpful, and I think articulating that a little further in the manuscript (together with the clarification in methods you described) will help make it easier for us to separate out the contribution here.
> >
> > > Indeed the suggestions you make regarding intrinsic reward for theorem-proving and targeting other forms of open-ended learning are interesting research avenues. These ideas should yield very fruitful research, though may be out of scope for our singular submission.
> >
> > Agreed! Future work.
> >
> > > Could you suggest some specific weaker operators that we could use for ablations?
> >
> > I was mostly thinking that simply seeing how it scales with weaker LLMs -- in each case, we'd have a sense of the baseline performance of the LLM without evolution, and what evolution adds. No need for a full scaling law...
> >
> > Overall, great work, and I appreciate the authors' clarifications. I'm raising my score to reflect this.

---

> > > ### Author Response · Authors · 2025-08-08
> > >
> > > Thank you again for the review! Your thoughts are very helpful in clarifying the setting of our approach and future work avenues! We will be sure to include all of this in the final version.

---

### Official Review · Reviewer_LEs3 · 2025-07-03

**Clarity:** 3
**Significance:** 3
**Originality:** 3
**Rating:** 4
**Confidence:** 4

**Summary:**

This paper presents two key contributions towards automating mathematical theory formation: FERMAT​​: A novel RL environment that models mathematical discovery as an MDP, with symbolic actions for concept discovery and theorem proving. EvoAbstract​​: An LLM-driven evolutionary algorithm that learns "interestingness" measures to guide theory exploration.

**Questions:**

1. Can you rely on stronger mathematical formalization systems such as Lean4?
2. What exactly are the external rewards? How are they designed?
3. Given that existing models possess a lot of prior knowledge, is there knowledge leakage when using the domain of number theory? Are there better domains for evaluation?

**Ethical Concerns:**

["NO or VERY MINOR ethics concerns only"]

**Final Justification:**

The author's response has solved my questions, and I will keep the score.

**Limitations:**

yes

**Quality:**

3

**Strengths And Weaknesses:**

Strengths:
1. The MDP formalization of theory formation is rigorous and opens new directions for RL research in mathematical discovery.

2. EvoAbstract's abstraction learning mechanism is a clever extension of evolutionary algorithms, enabling compositionality in program synthesis.

Weaknesses：
1. Evaluated only on elementary number theory; generalization to other mathematical domains remains unproven.

2. Dependence heavily on Z3

---

> ### Author Rebuttal · Authors · 2025-07-31
>
> Thanks for your review of our paper! We appreciate your comments on the significance of the posing of the theory formulation problem and the evolutionary approach! We respond to your comments and questions below. Happy to discuss more, please share any other questions you have!
> > Evaluated only on elementary number theory; generalization to other mathematical domains remains unproven.
>
> **Response**: This is a great point! We agree that another domain would help illustrate the benefits of our method and provide another off-the-shelf exploration point for the community. To demonstrate the extensibility of the FERMAT framework for our rebuttal, we have implemented the theory of Galois fields. We chose this domain for its rich structure that is applicable to a variety of mathematical fields, from coding theory to combinatorial design, which human mathematicians have only developed in the past 200 years. Like in elementary number theory, we have curated a list of 50 ground truth entities for this domain, to be included in the revision.
>
> For verification in this domain, we leverage the finiteness of Galois fields by implementing a simple mechanism that exhaustively checks the validity of a predicate over all field elements, when feasible. In principle, this approach could be replaced by a more sophisticated prover (e.g. some finite-field prover in Lean).
>
> We reran the baseline experiments (with lesser compute of 64 iterations with 30 seconds to an episode due to the time constraint) and confirmed that our evolutionary approach learns interestingness functions that outperform fixed HR measures.
> Among hardcoded metrics, comprehensibility performed best with a reward of 4.48, while equal weighting scored 3.00. Funsearch found a measure scoring 6.52, while our method, EvoAbstract, achieved 7.78, slightly ahead of Funsearch. Here is a complete table of our experiments.
> **Finite Fields added (F_27 in experiments)**
> **# Ground Truth Entities:** 50
> **Most interesting ground truth entities:** primitivity, order, characteristic
>
> | Heuristic/Method               | Mean    | Stdev   |
> |--------------------------------|---------|---------|
> | Zero weight/Random (64 eps)    | 2.0968  | 1.2915  |
> | Novelty (64 eps)               | 2.2295  | 1.2200  |
> | Parsimony (64 eps)            | 1.9516  | 1.4077  |
> | Applicability (64 eps)        | 2.8125  | 1.2732  |
> | Comprehensibility (64 eps)    | 4.4844  | 1.7138  |
> | Equal weight (64 eps)         | 3.0000  | 1.3346  |
> | Productivity (64 eps)         | 2.5397  | 1.4889  |
> | GPT-4o                        | 2.7413  | 0.8718  |
> | GPT-4o (best)                 | 4.9839  | 1.9633  |
> | Funsearch (64 iter)           | 6.5156  | 3.4003  |
> | EvoAbstract (64 iter)         | 7.7812  | 3.8910  |
>
> We plan to include more experiments with this domain in our next revision, especially to measure the variance of evolution runs.
>
>
> > Dependence heavily on Z3, Can you rely on stronger mathematical formalization systems such as Lean4?
>
> **Response**: There is actually **no fundamental dependence on Z3**. We implement everything in the form of a generic DSL and cross-compile our DSL to Z3. The critical purpose of our DSL is to support nested definitions, which allows for modular construction of definitions and conjectures. The DSL is designed such that it can also be cross-compiled to other languages like Lean. Details on this are reported in Appendix A.3 (page 16, lines 647 - 669), which also discusses how we include a specific instantiation for compilation into SMT-LIB format. This was chosen for simplicity as we can use Z3 as a black box prover which allows us to focus on the less-explored problems of conjecturing and producing definitions. There is nothing that prevents using Lean, for example, inside of Fermat. However, using Lean introduces an extra complexity, as one needs to produce a suitable prover which is another moving part. Such a prover would likely have to learn to prove from scratch which is another difficult problem worthy of an entire field of research [1]. The generic DSL and compiler we wrote does the heavy lifting for composing definitions, which is not readily available in any language. We are happy to describe the specific details of how it works.
>
> To summarize, Z3 is not too coupled to Fermat, it just serves as a useful prover tool that helps us isolate the definition-making and conjecturing task. As we mention in the conclusion, extensions to Lean are very easy to produce in Fermat. The hard part of scaling to more complex mathematics is to produce a method for learning to prove (from low data), which is a very promising research direction. Let us know if you have any questions about this!
>
> > What exactly are the external rewards? How are they designed?
>
> **Response**: Thanks for this question! The external reward is a measure of how much an automatically generated theory intersects with a ground-truth set of human-made mathematical objects. This is because we are ultimately interested in learning how to classify objects that humans find interesting. We take a set of human-made mathematical objects (primality, square numbers, etc.) from a number theory textbook to serve as the ground truth reward. Information on this can be found on lines 205 - 214 and in Appendix A.4. The full ground truth benchmark is available in the supplementary material as well! Let us know if you have any questions!
>
> > Given that existing models possess a lot of prior knowledge, is there knowledge leakage when using the domain of number theory?
>
> **Response**: Good question! The way we use the LLM is separate from the domain we study. The LLM only sees previous interestingness measures (as a Python program) and a numerical score measuring how many ground-truth concepts (on average) that measure produces during theory formation. It never sees any concepts that get produced, and the model is never even *told* that it is even working on number theory. You can inspect the prompt on page 24 (Figure 12) in the Appendix (and in the supplementary material) to confirm this. In this way, there is unlikely to be any number-theory leakage through our experiments.
>
> Let us know if we can answer any further questions!
>
> [1] Tim Gowers, How can it be feasible to find proofs? (2023)

---

> > ### Comment · Reviewer_LEs3 · 2025-08-08
> >
> > The author's response has solved my questions, and I will keep the score.

---

### Official Review · Reviewer_uroY · 2025-07-03

**Clarity:** 3
**Significance:** 3
**Originality:** 3
**Rating:** 5
**Confidence:** 2

**Summary:**

The paper presents a reinforcement learning environment called Fermat which is subsequently used to learn “interestingness” of mathematical objects. The problem was modeled as intrinsic reward optimization where the goal is to discover the best intrinsic reward function I that can guide action selections to maximize the long-term (sparse) task specific reward. The intrinsic reward functions are implemented as python programs that can be generated by LLMs. More specifically, the authors prompts one LLM () to propose candidates as if it’s doing evolution search, and use another LLM to identify useful subroutines from evolved programs, which are provided as additional context to the first LLM.  The authors show that the learned interestingness measures can better guide the discovery of ground-truth mathematical entities in the theory of arithmetic and its simple fragment.

**Questions:**

1. Clarification needed: page 19, algorithm 2, the interesting measure I takes an entity and the graph G, but by definition on lines 80 and 94 it should take states and actions. Is there a mix of notations here?
2. In the discussion you mentioned that the templates limited search space, and I agree it is a good idea to do so to limit the computational resources required. I’m just curious, do you have preliminary results when you don’t provide a template at all? What’s the behavior of the LLMs in that case?

**Ethical Concerns:**

["NO or VERY MINOR ethics concerns only"]

**Final Justification:**

I am happy with the discussion and the philosophical insights. I'd like to keep my positive assessment.

**Limitations:**

Yes.

**Paper Formatting Concerns:**

The formatting seems proper.

**Quality:**

3

**Strengths And Weaknesses:**

Strength
- The overall design of the system is interesting. In particular, instead of blatantly asking the LLM to criticize itself, the integration of an evolution-like algorithm to update the programs proposed by the LLM provides more factual groundings for candidates selection. In addition to that, the abstraction learning also makes sense because validated sub-routines help the (first) LLM avoid going wild, thereby guiding the search towards more promising regions of the program space.
- The careful formalization makes the problem setting clear. The production rules with examples really helped with understanding how things are modeled as an MDP and how the system progresses. The overall presentation is very good.
- The qualitative analysis (in addition to the quantitative one) provides explanatory insights. Since what’s generated by the proposed approach are explicit python programs, the exact behavior of them can be observed and interpreted. The authors demonstrated that some subroutines do look similar to human-engineered measures proposed by previous works.

Weakness
- I am somewhat doubtful about the branding of “interestingness”. My understanding is that the system is essentially just learning the intrinsic reward functions (given the sparse extrinsic one), but this function does not necessarily express interestingness. A transition step can be given a high reward because it’s a tedious but necessary construction towards the target entity. As to the real interestingness, I think you have already “explicitly” defined it using the set of entities \mathcal{E} (which induces the extrinsic reward)?
- Minor: learning an intrinsic reward function is fine, but if re-constructing well-known interesting mathematical entities is the ultimate goal, maybe other regular techniques in RL that can deal with sparse rewards are also worth trying.

---

> ### Author Rebuttal · Authors · 2025-07-31
>
> Thank you for the thorough review of our paper! We appreciate that you found the presentation clear and that the research methods are sensible and interesting. We respond to your comments and questions below (and are happy to engage in discussion about our work!):
>
> > I am somewhat doubtful about the branding of “interestingness”. My understanding is that the system is essentially just learning the intrinsic reward functions (given the sparse extrinsic one), but this function does not necessarily express interestingness. A transition step can be given a high reward because it’s a tedious but necessary construction towards the target entity. As to the real interestingness, I think you have already “explicitly” defined it using the set of entities \mathcal{E} (which induces the extrinsic reward)?
>
> **Response**:  You raise a good point, and your understanding of the system is accurate. We claim that if the agent is able to learn a measure which can successfully reproduce many entities from the ground truth set, then this measure can be thought of as measuring interestingness because the ground truth set consists exactly of entities which humans find interesting. Our effort is towards identifying a concise classification of these human-made mathematical objects.
>
> In general, the definition of interestingness has be long-debated and somewhat ill-defined [1-4]. Some properties mathematicians [1,2] have discussed value mathematical objects if they somehow connect disparate fields, or their proofs unveil a deep complexity that is surprising in some way. Informational theoretical approaches [3] contend that interesting mathematical statements, that is, the ones humans have developed, are valuable because they form a ¨representative core¨ of all mathematical statements. Another perspective can be gleaned from [Wigner, 4] that mathematical objects are interesting because of their uncanny ability to model real-world phenomena. Our approach generally aligns with all these philosophies. For instance, our simple novelty baseline metric is intended to measure how surprising a result is, based on what is known, and comprehensibility seeks to reward simpler-stated. Further, our main contention is that if a measure leads to reproduction of many human-made concepts, then that measure ought to be well-aligned with the ¨true¨ interestingness measure that produced these concepts. We like your point about tedious constructions leading to interesting concepts, though we think that a tedious construction should be dubbed interesting because of its utility in defining interesting concepts later on. This rationale is similar to recursive measures of interestingness that were developed in AM (1977). We would be happy to discuss more about this topic, the history is very rich and we would love to clarify where our approach stands amongst historical opinions! We will also be including these discussions in our next revision.
>
> > Minor: learning an intrinsic reward function is fine, but if re-constructing well-known interesting mathematical entities is the ultimate goal, maybe other regular techniques in RL that can deal with sparse rewards are also worth trying.
>
> **Response**: We agree that this is an excellent avenue for future research, and it is plausible that these could scale to developing more interesting theories. However, we also think having a Python program as a kernel for characterizing interesting objects lends itself to interpretability. Other methods may also be less interpretable in understanding *why* human mathematics is interesting.
>
> > Clarification needed: page 19, algorithm 2, the interesting measure I takes an entity and the graph G, but by definition on lines 80 and 94 it should take states and actions. Is there a mix of notations here?
>
> **Response**:  Thank you again for the extremely thorough inspection of our paper. You are right to point this out, as there is a mix of notations here. We mean to distinguish the overall policy, which can be thought of as a function which measures the interestingness of actions (by providing a probability distribution over actions given a state), with the measure we learn that assigns an interestingness score to the entities in the graph. These two problems are similar up to the policy template which is a necessary scaffolding to make use of the interestingness scores of the entities. We agree that this mix of notations can be somewhat confusing and will clarify in our next revision.
>
> > In the discussion you mentioned that the templates limited search space, and I agree it is a good idea to do so to limit the computational resources required. I’m just curious, do you have preliminary results when you don’t provide a template at all? What’s the behavior of the LLMs in that case?
>
> **Response**: Great questions! The template serves as a means to scaffold the entity scorer to produce a policy. This is done because we have base measures like comprehensibility and novelty, which operate directly on attributes of the entity instead of on any information about an action. Because of this, removing the template entirely will not work because there is no mechanism that scores actions independently, as the LLMs are only involved in producing the python program which scores the entities. However, it is a really interesting question to try to incorporate information about actions into the learned Python program, though this will require additional mechanisms for scoring production rules independently. Similarly, the evolution phase need not only be involved in producing the Python program, but the policy template itself is an evolvable piece of code. We think these are both promising future research directions that should enable better scaling for theory exploration.
>
> Let us know if you have further questions! We would be happy to discuss more.
>
>
> [1] A Mathematician’s Apology (G. H. Hardy, 1940)
>
> [2] Mathematical Creation (Henri Poincare, 1908)
>
> [3] Machine Learning and Information Theory Concepts towards an AI Mathematician (Bengio, Malakin, 2024)
>
> [4] The Unreasonable Effectiveness of Mathematics in the Natural Sciences (Wigner, 1960)
>
> [5] AM: An Artificial Intelligence Approach to Discovery in Mathematics as Heuristic Search (Lenat, 1977)

---

> > ### Comment · Reviewer_uroY · 2025-08-06
> >
> > Thank you for the response. I am happy with the discussion and the philosophical insights, though personally I have a different point of view (which does not affect my evaluation of this submission). I'd like to keep my positive assessment.

---

### Official Review · Reviewer_zbMW · 2025-07-03

**Clarity:** 2
**Significance:** 2
**Originality:** 3
**Rating:** 4
**Confidence:** 3

**Summary:**

In this work, the authors introduce a new reinforcement environment called FERMAT for mathematical discovery. In particular, the mathematical theory is modeled as a Markov Decision Process (MDP), where the state is a graph representing mathematical entities (vertices) and dependencies between the entities (edges). The key challenge of the paper is how to automatically score the "interestingness". To do this, the authors develop an approach called EvoAbstract, which is LLM-based, with the aim of learning the intrinsic value of a mathematical object. The evaluation is done using the elementary number theory, using the Z3 solver as the prover.

**Questions:**

- To what extent can this approach be translated and used with another prover, e.g., if Rocq, Lean, or HOL4 is used instead? That is, how tightly integrated is the approach to Z3?

- In Section 5.1, it says  "However, agents guided by the best discovered measures can still prescribe high interestingness to entities of little value." So, what does this mean? If a system prescribes high interestingness value, and you (as a human) still think it is little value, why is a certain interestingness score good?

- To what extent is the "interestingness"  biased on the LLM's knowledge of interesting theorems (i.e., theorems that have been part of the training data for the LLM)? To what extent can we then discover new theorems of interest that have not already been discovered?

- It is not completely clear how the actions for proving works. t says on line 68 "attempts to prove or discover it". But how is the action of proving or disproving a theorem then captured in the MDP? Please clarify this, if possible.

**Ethical Concerns:**

["NO or VERY MINOR ethics concerns only"]

**Final Justification:**

Thank you for your comprehensive clarifications and explanations. In particular, I liked your discussion about interestingness. Please make sure to include this in the final version of the paper. I also appreciate that you have included Galois fields.

It sounds promising that the approach is not limited or specific to Z3. However, to actually show that it works in dependently-typed interactive theorem provers, such as Lean and Rocq, further work is, of course, needed. I agree that this is non-trivial, but an important and interesting future direction. Given the rebuttal and also reading the other rebuttals and reviews, I have decided to increase my score.

**Limitations:**

Limitations are briefly discussed in the discussion section.

**Paper Formatting Concerns:**

No formatting problem

**Quality:**

2

**Strengths And Weaknesses:**

### Strengths ###

- The paper addresses an interesting and relevant problem of mathematics, working in a more open-ended process

- The paper is fairly well-written and clearly motivated, although the central idea concept of interestingness is less clear.

- The approach of defining the mathematical problem as an MDP and then using reinforcement learning to discover theorems and conjectures is an interesting research direction.


### Weaknesses ###

- The theory that is used as evaluation is quite limited, and the work does not explore or evaluate the use of richer theories. This is also one of the weaker parts of the paper; by using a prover such as Z3, there are limitations of what can be achieved, compared to if a higher-order logic (e.g., HOL4) or dependently type theory (e.g., Lean or Rocq) is used.

- I find the description of the central concept of "interestingness" to be rather vague. As a consequence, the performance comparison with manual interestingness (as described in Section 5) is not obvious. I would suggest that a deeper discussion about interestingness is added to the paper, which includes examples of different levels of interestingness.

- Some figures need clearer explanations. For instance, Figure 2 (which is important) can be described in a clearer way. One way of doing it is to add numbers to the figure and then describe the different parts step-by-step, referring to these numbers in the figure in the caption.

---

> ### Author Rebuttal · Authors · 2025-07-31
>
> Thank you for your detailed review of the paper! We respond to your comments and questions below, and would love to discuss further during the discussion period.
>
> > The theory that is used as evaluation is quite limited, and the work does not explore or evaluate the use of richer theories.
>
> **Response**: We agree that more complex theories are interesting avenues for study! It was indeed the intention to begin with elementary number theory, and then to expand to richer theories in future work. The FERMAT framework is capable of such extension because it does not inherently rely on either the domain or the prover (in this case Z3), and can be readily adapted for more complex theories and theorem provers. We chose Z3 for its reliability, allowing us to focus on the core challenges of the paper - definition and conjecture synthesis - while treating learning to prove as an orthogonal and very important direction for future work.
>
> To demonstrate the extensibility of the FERMAT framework for our rebuttal, we have implemented the theory of Galois fields. Like in elementary number theory, we have curated a list of 50 ground truth entities (including primitivity, order, characteristic, and lines) for this domain, to be included in the revision.
>
> We reran the baseline experiments (with lesser compute of 64 iterations with 30 seconds to an episode due to the time constraint) and confirmed that our evolutionary approach learns interestingness functions that outperform fixed HR measures.
> Among hardcoded metrics, comprehensibility performed best with a reward of 4.48, while equal weighting scored 3.00. Funsearch found a measure scoring 6.52, while our method, EvoAbstract, achieved 7.78, slightly ahead of Funsearch.
> We plan to include more experiments with this domain in our next revision, especially to measure the variance of evolution runs in this domain.
>
> > To what extent can this approach be translated and used with another prover, e.g., if Rocq, Lean, or HOL4 is used instead? That is, how tightly integrated is the approach to Z3?
>
> **Response**: This approach is easily extensible to interactive theorem provers. Our implementation is **not specific to Z3**. The way we’ve designed the framework is to use a general DSL for composition of definitions that can be cross-compiled to multiple languages like Z3, Lean, Rocq, etc. Since we want to support multiple target languages while implementing all the production rules, creating such a DSL becomes nontrivial because we need to keep track of nested variable scopes needed in our composed definition. Despite all the challenges involved in designing the DSL itself, we built a compiler for our DSL so that it can be transpiled to Z3 (and can be extended to other languages in the future).
>
> However, writing proofs in Lean is nontrivial and introduces significant complexity which can distract from the exploration of learning how to produce definitions and conjectures in easy-to-describe theories. Trying to learn a Lean prover from low data in itself is a central research question which is at the heart of AI.
>
> That being said, we completely agree that extensions to more expressive languages is a natural next step in this research direction.
>
> > I find the description of the central concept of "interestingness" to be rather vague. As a consequence, the performance comparison with manual interestingness (as described in Section 5) is not obvious. I would suggest that a deeper discussion about interestingness is added to the paper, which includes examples of different levels of interestingness.
>
> **Response**: Thank you for this valuable suggestion. Formally defining mathematical interestingness is a profound challenge that has been explored by mathematicians for over a century. We plan to incorporate a thorough treatment in our next revision, as its inclusion will clarify the motivation for our approach and the interpretation of our results.
>
> In plain speech, the core of “interestingness” is the ability to discern what mathematical concepts are “worth pursuing”, which we believe is critical for self-directed discovery of math, yet rarely addressed. Interestingness is the scientific potential to explore further. Human mathematicians naturally find themselves “drawn” to certain topics, whether for fixation on a certain problem, or for aesthetic appreciation, or other idiosyncratic reasons. This allows the mathematician to narrow down the combinatorially large space of mathematical exploration. For an AI agent to discover new math, it would need a similar mechanism to navigate the huge search space of mathematics. This is our core reason for trying to **learn** interestingness, which we believe to be the crucial direction in AI for math.
>
> There are many views on what makes mathematics interesting. Poincare [2] saw beauty in the harmonious connection of seemingly unrelated ideas, while Hardy valued depth and generality over practical use. We also believe simple mathematics can be interesting through its real-world applicability [4].
>
> The most important takeaway of our performance comparison in section 5 is the ability to programmatically, iteratively **learn** such an interestingness measure to direct math discovery, as opposed to hardcoded heuristics that do not capture the complexity of theory formation. In this section, our aim is to show that we can already do significantly better than such heuristics, and that further work can lead us to still further improvements. The end goal would be to have a learned interestingness advanced enough to truly discover new math.
>
> Finally, as per your suggestion, we agree that different levels of interestingness would be a helpful illustration. Here is a rough example, a version of which we would add to the paper:
>
> Current highest level (human mathematician level): I want to study the symmetries of certain groups because I think it will prove the unsolvability of the quintic.
>
> AI level: I want to make more concepts using equality because this concept has certain properties within the graph of discovered mathematical concepts (depth, connectivity, age, arity) that I have learned to prioritize with certain weightings.
>
> Heuristic level: I want to make more concepts using equality because I always prioritize concepts with the highest connectivity.
>
> No built-in notion of interestingness: Random policy
>
> Let us know if you have questions or comments about this discussion! It is at the heart of our effort and we will be sure to include it in our next revision.
>
> > In Section 5.1, it says "However, agents guided by the best discovered measures can still prescribe high interestingness to entities of little value." So, what does this mean? If a system prescribes high interestingness value, and you (as a human) still think it is little value, why is a certain interestingness score good?
>
> **Response**: Good question! The main thing that we point out there is that even the best performing interestingness metrics can cause the agent to get stuck in a local minimum - where it overvalues a concept while neglecting other concepts. In the case of our best run, the agent assigns the equality concept a very high score, which causes the algorithm to focus on developing equality related concepts at the expense of discovering new things. It is also the case that certain ground truth concepts are discovered, but subsequently not interesting enough to be further expanded (as we mention in the paper, primality is a good example). This is all to say that the interestingness measures we find are relatively good, but certainly not optimal.
>
> > To what extent is the "interestingness" biased on the LLM's knowledge of interesting theorems (i.e., theorems that have been part of the training data for the LLM)? To what extent can we then discover new theorems of interest that have not already been discovered?
>
> **Response**: Great questions! In our experiments, the LLM is never informed of the internal operation of Fermat and *never sees any definitions or conjectures created*. It does not even know that the domain is elementary number theory! The only signal to the LLM is 1. The program which implements the measure and 2. A numerical score obtained by using that measure for theory exploration. In this way, the LLM can’t use its existing knowledge about math to manipulate the objects produced during theory exploration, because they are never provided. Further, the LLM is never instructed that it is even working on the number theory domain in the first place. The LLM’s purpose is to provide an interestingness kernel that guides a relatively open-ended agent, so we see the discovery of new theorems of interest as entirely plausible by scaling this research effort.
>
> > But how is the action of proving or disproving a theorem then captured in the MDP? Please clarify this, if possible.
>
> Thanks for pointing this out! We mentioned on lines 74-75 that if a conjecture is proven, the state removes the entity as a conjecture and replaces it with a theorem, and a proof object given by Z3. We didn’t mention what happens when a conjecture is disproved, which is that the conjecture receives a disproved status and a counterexample is attached to it. We’ll be sure to add this detail in our revisions, and explicitly describe what transition function \delta_prove governs the proving behavior in the MDP.
>
> > For instance, Figure 2 (which is important) can be described in a clearer way…
>
> This is a really nice suggestion, thank you! We will incorporate it in our revisions.
>
> Thank you again for your thorough consideration of our paper! We would be happy to answer any further questions you might have!
>
>
> [1] A Mathematician’s Apology (G. H. Hardy, 1940)
>
> [2] Mathematical Creation (Henri Poincare, 1908)
>
> [3] Machine Learning and Information Theory Concepts towards an AI Mathematician (Bengio, Malakin, 2024)
>
> [4] The Unreasonable Effectiveness of Mathematics in the Natural Sciences (Wigner, 1960)

---

> > ### Comment · Reviewer_zbMW · 2025-08-04
> >
> > Thank you for your comprehensive clarifications and explanations. In particular, I liked your discussion about interestingness. Please make sure to include this in the final version of the paper. I also appreciate that you have included Galois fields.
> >
> > It sounds promising that the approach is not limited or specific to Z3. However, to actually show that it works in dependently-typed interactive theorem provers, such as Lean and Rocq, further work is, of course, needed. I agree that this is non-trivial, but an important and interesting future direction. Given the rebuttal and also reading the other rebuttals and reviews, I have decided to increase my score.

---

> > > ### Author Response · Authors · 2025-08-08
> > >
> > > Thanks again for the consideration of our paper! We will make sure to include the discussion about interestingness in the final version!

---

### Note · Authors · 2025-08-13

We would like to thank the reviewers for their consideration of our paper and useful discussions that helped clarify the message of our work.

We found the main points discussed to be very helpful:

1. Fermat is not specific to Z3. Our framework contains a generic DSL which supports cross-compilation to other languages. We use Z3 as a specific instantiation so that it serves as a block-box prover so we can better understand the definition-making and conjecturing task inside of theory formation. We will clarify this in the main body of the paper in our next version.
2. Through discussions with the reviewers we have clarified that the definition of interestingness has been historically debated, and in attempts to define it in prior theory formation efforts, our work is a natural successor. We also really appreciated the suggestions from Reviewer RkzN which were useful in placing our effort in the context of Intrinsically Motivated and Meta-RL. We plan to elaborate on the historical context of interestingness and these topics, as we mentioned in our rebuttals, in our next version.
3. An additional domain: We included experiments with Galois Fields and demonstrate that the evolutionary approach obtains results consistent with the findings in elementary number theory. We plan to include these experimental results in our next version.

Thanks again to the reviewers, and for the AC for coordinating the review process on this submission. We will be sure to include these clarifications in our next revision, along with style changes and readability improvements.

---

### Decision · Program_Chairs · 2025-09-17

**Decision:**

Accept (spotlight)

**Comment:**

The context of the paper is that of RL-based concept discovery and theorem proving, building upon/extending Lenat's AM and Eurisko systems.
An evolutionary approach is proposed to learn an "interestingness function" (IF), viewed as a way to narrow down the combinatorially large space of mathematical exploration.

This IF is successfully compared to hardcoded heuristics. Note that, as could be expected, the proposed IF is not optimal (can get the MDP stuck in uninteresting regions).

Reviewers are generally very positive about the approach:

zbMW and LEs3 had concerns with the generality of the approach; authors successfully addressed this concern,
implementing an extension including the theory of Galois fields.

uroY finds the overall presentation very good, with explanatory insights. Still, a question is whether an "interestingness function" is equivalent to a standard RL intrinsic reward.

RkzN asks to which extent the good performance of the approach depends on the IF initialization (currently derived from GPT-4o, a rather strong baseline).

c2rm suggests that using other LLMs (e.g., Qwen, Gemini) could serve as alternative. The authors agree and mention that the cost is affordable.
The AC is curious whether the use of other LLMs has an impact on the IF.